# Intricacies of human–AI interaction in dynamic decision-making for precision oncology

Dipesh Niraula [1] ✉, Kyle C. Cuneo [2], Ivo D. Dinov[3], Brian D. Gonzalez[4], Jamalina B. Jamaluddin [5], Jionghua Judy Jin[6], Yi Luo [1], Martha M. Matuszak[2], Randall K. Ten Haken[2], Alex K. Bryant[2,7], Thomas J. Dilling [8], Michael P. Dykstra[2], Jessica M. Frakes[8], Casey L. Liveringhouse[8], Sean R. Miller[2], Matthew N. Mills[8], Russell F. Palm [8], Samuel N. Regan[2], Anupam Rishi[8], Javier F. Torres-Roca[8], Hsiang-Hsuan Michael Yu[8] & Issam El Naqa [1] ✉

AI decision support systems can assist clinicians in planning adaptive treatment strategies that can dynamically react to individuals' cancer progression for effective personalized care. However, AI's imperfections can lead to suboptimal therapeutics if clinicians over or under rely on AI. To investigate such collaborative decision-making process, we conducted a Human–AI interaction study on response-adaptive radiotherapy for non-small cell lung cancer and hepatocellular carcinoma. We investigated two levels of collaborative behavior: model-agnostic and model-specific; and found that Human–AI interaction is multifactorial and depends on the complex interrelationship between prior knowledge and preferences, patient's state, disease site, treatment modality, model transparency, and AI's learned behavior and biases. In summary, some clinicians may disregard AI recommendations due to skepticism; others will critically analyze AI recommendations on a case-by-case basis; clinicians will adjust their decisions if they find AI recommendations beneficial to patients; and clinician will disregard AI recommendations if deemed harmful or suboptimal and seek alternatives.

Development of novel cancer therapies[1,2] and increasing availability of longitudinal multi-omics data[3,4] have improved our ability to prescribe personalized, adaptive treatment strategies[5–7] capable of dynamically reacting to individuals' cancer progression while improving efficacy and minimizing side effects. However, the dynamic nature of adaptive strategies compounded by a wide range of clinical options, high data dimensionality, uncertainty in assessing treatment response, and the uncertainty in the future course of disease, present challenges in tailoring optimal strategies[7]. Assistance from artificial intelligence (AI) decision-support tools designed for precision oncology[8–12] that can provide individual treatment response assessments, outcome predictions, and optimal treatment recommendations can overcome such challenges. However, AI tools are not perfectly accurate, have inherent biases, and are limited by the quality of their training data[13,14]; as such,

[1]Department of Machine Learning, Moffitt Cancer Center, Tampa, FL, USA. [2]Department of Radiation Oncology, University of Michigan, Ann Arbor, MI, USA. [3]Department of Health Behavior and Biological Sciences, University of Michigan, Ann Arbor, MI, USA. [4]Department of Health Outcomes and Behavior, Moffitt Cancer Center, Tampa, FL, USA. [5]Department of Nuclear Engineering and Radiological Sciences, Moffitt Cancer Center, Tampa, FL, USA. [6]Department of Industrial and Operations Engineering, University of Michigan, Ann Arbor, MI, USA. [7]Department of Radiation Oncology, Veterans Affairs Ann Arbor Healthcare System, Ann Arbor, MI, USA. [8]Department of Radiation Oncology, H. Lee Moffitt Cancer Center & Research Institute, Tampa, FL, USA. ✉e-mail: dipesh.niraula@moffitt.org; issam.elnaqa@moffitt.org

over/under reliance on AI[15] can result in sub-optimal therapeutics. Therefore, investigating collaborative Human-AI decision-making behavior[16,17] is crucial before treating advanced diseases that have a relatively narrow therapeutic window and tighter margin of error, such as cancer[9,16–20].

We conducted a Human–AI interaction study to investigate clinicians' (physicians and residents) collaborative decision-making behavior in knowledge-based response-adaptive radiotherapy (KBR-ART)[21–30]. KBR-ART is an actively researched single-modality and single-intervention adaptive treatment strategy designed to improve patient outcomes. It consists of three phases: pre-treatment assessment, response evaluation, and adaptation. In the response evaluation phase, patients' treatment response is assessed by comparing pre and during-treatment multi-omics information, and in the adaptation phase, a treatment plan is adapted (dose escalation/de-escalation). In this study, the clinicians collaborated with ARCliDS[22,31,32]—a software for dynamic decision-making developed using a model-based deep reinforcement learning algorithm. For application in KBR-ART, ARCliDS uses a graphical neural network-based model of radiotherapy environment which defines a patient's state via a graph of multi-omics features and is capable of assessing treatment response and predicting treatment outcomes. Prior to this study, we had developed ARCliDS modules for two modalities that were trained on two retrospective cohorts: non-small cell lung cancer (NSCLC) patients who had received adaptive radiotherapy (RT)[29] and hepatocellular carcinoma (HCC) patients who had received adaptive stereotactic body radiotherapy (SBRT)[30]. In this study, we designed a two-phase Human–AI interaction study for each of the two modules, in which clinicians were asked to prescribe mid-treatment dose adaptation without (Unassisted phase, Human Only) and with AI-assistance (AI-assisted phase, Human + AI) for a number of retrospective patients. In addition, they were asked to input their decision confidence level, and trust level on AI recommendation, and were encouraged to provide text remarks.

We designed web modules that closely simulate KBR-ART's decision-making process, as summarized in Fig. 1. In the Unassisted phase, we presented RT treatment plan (dose volume histogram, and three-dimensional dose distribution) and CT/PET/MRI imaging from KBR-ART's response evaluation phase, and asked clinicians to assess mid-treatment response and input their dose decision for the remaining of treatment period along with their confidence level (0-5, 5 being the highest level). In the AI-assisted phase, we provided ARCliDS recommendations based on its assessment of treatment response and asked the evaluators to re-enter their decision and decision confidence level. Besides AI recommendation we provided additional graphs to improve model transparency and explainability for establishing and maximizing trust on AI[9,16,17,20,33]. We provided a two-dimensional outcome space for quantifying tradeoffs between tumor control probability (TCP) and normal tissue complication probability (NTCP)[34] across a range of RT dose fractionation options; graphically demonstrated the model uncertainty for both AI recommendation and outcome prediction; and provided feature distribution plots with feature value marked in the foreground for presenting the "whereabouts" of the individual patient with respect to the rest of the population. To assess evaluators' level of trust in the individual AI recommendation (0-5, 5 being the highest level; AI Trust Level), we included four multiple-choice questions. Lastly, we provided text boxes for remarks to gain insights into the collaborative decision-making process.

The main objectives of this study were twofold: broader model-agnostic investigation of collaborative decision-making process, and model-specific application-grounded evaluation[35] of ARCliDS by domain expert and end-users. We included diverse study elements for attaining both levels of our objectives. We incorporated AIs for two diseases and two treatment modalities and enrolled evaluators from two medical institutions located in different states, with different experience levels, sub-specialties, and professional backgrounds.

Considering the fact that evaluators' mental model of AI can affect human-AI interaction[36], we took the following steps to homogenize evaluators' first impression of AI: (i) created two 10-minute tutorial videos[37,38]; (ii) conducted a pre-evaluation information session with the evaluators, in which, we played the training video and demonstrated sample evaluation followed by a question-and-answer round; and (iii) included a web-link to the tutorial video and original ARCliDS manuscript[39] in the evaluation modules. Furthermore, to optimize the evaluator's time utilization, we designed the evaluation to take no more than an hour, added a user account system so the evaluation could be completed in multiple sessions if needed, and enabled automatic saving of all user inputs in the cloud.

Here, we found that AI-assistance does not homogeneously influence all experts and clinical decisions. Evaluators will adjust their decision closer towards AI recommendation only if they agree with that recommendation and only, if they find it necessary. The underlying differences in disease type and treatment modality can result in differences in collaborative behavior. AI-assistance can decrease inter-physician variability. Evaluators are generally more confident in making decisions close to the standard clinical practice. Analysis of evaluators' remarks indicated concerns for organs at risk and RT outcome estimates as important decision-making factors.

## Results

### AI-assistance did not homogeneously influence all decisions

Our study accumulated 72 evaluations for NSCLC cohort (9 evaluators × 8 patients) and 72 evaluations for HCC cohort (8 evaluators × 9 patients). First, we analyzed AI-influence on a distribution level by performing a matched paired randomization t-test[40,41] between unassisted decision ($un$) and AI-assisted decision ($aia$). As shown in Fig. 2a, b, e, and f, we failed to reject the null hypothesis in most of the tests, grouped by both evaluators and patients, indicating the absence of significant AI influence. However, on an individual level, we observed that AI assistance resulted in the adjustment of about half of the decisions [41(57%) for NSCLC, and 34(47%) for HCC], which is a considerable proportion. Moreover, we found only 2 out of 17 evaluations contained zero decision adjustments i.e.,$un = aia$ for all cases, while the remaining 15 evaluations contained at least 2 decision adjustments [88% (15/17)]. In particular, NSCLC evaluators #7 and #8 made zero decision adjustments, however, all HCC evaluators made at least two decision adjustments. As shown in Fig. 2c, d, number of NSCLC decision adjustments ranged from 0% to 100% among evaluators and 44% to 78% among patients and as shown in Fig. 2g, h, number of HCC decision adjustments ranged from 22% to 67% among evaluators and 25% to 75% among patients. Decision adjustment level (Gy/fx) is summarized in Supplementary information. In addition, to investigate the relationship between unassisted and AI-assisted decisions, we performed a correlation analysis on clinical decisions, as presented in Supplementary information. We found a statistically significant positive correlation between $un$ and $aia$ but the correlation coefficient was not strong enough (i.e., near 1) to dismiss AI influence, further corroborating the observation that AI influence exists but not all experts and not all clinical decisions are influenced homogeneously.

### Decision adjustment positively correlated with dissimilarity to AI

Under a positive AI influence, we expect evaluators to adjust their decision closer towards AI recommendation ($ai$). In such a case, we expect higher level of decision adjustment ($aia - un$) for higher level of dissimilarity between AI recommendation and unassisted decision ($ai - un$) and vice versa. Conversely, we would expect no decision adjustment, $aia = un$, when, $ai = un$. As expected, we found an overall positive correlation between decision adjustment level and dissimilarity in decision-making with AI as shown in Fig. 3a–c. Specifically, for NSCLC, $\rho = 0.53(p < 0.001)$, for NSCLC when excluding evaluators with

# ARCliDS Evaluation Module Workflow

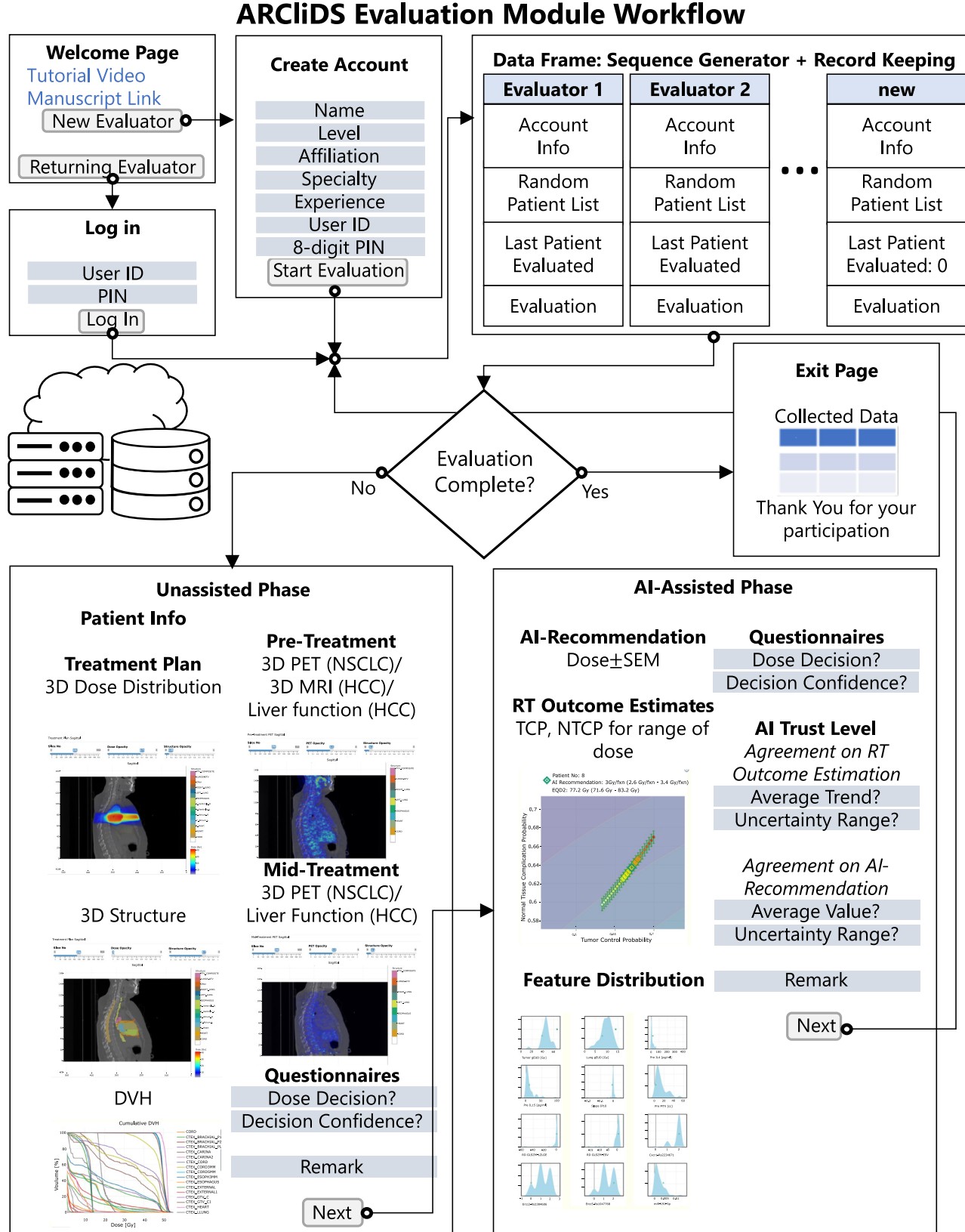

zero decision adjustment, $\rho = 0.58(p<0.001)$, and for HCC, $\rho = 0.60(p<0.001)$. The positive spearman correlation coefficient indicates an increasing monotonic relationship between $ai - un$ and $aia - un$. Here, we separately investigated the NSCLC cohort excluding zero decision adjustment to examine the contribution of strictly positive AI influence. Additionally, we observed that the majority of the data points reside in the first and third quadrant, which represents decision adjustments in alignment with AI's advice, whereas very few data points reside in the second and fourth quadrants which represents decision adjustment against AI's advice. These results further corroborate a positive AI influence on clinicians' decisions.

**Fig. 1 | ARCliDS evaluation module workflow.** The modules were deployed on cloud in shinyapps.io server and used google sheets as data storage. Welcome Page contains links to tutorial video hosted in YouTube and manuscript preprint; from which a new evaluator can create a new account and returning user can login back to complete their evaluation. Create Account page consists of a series of input prompts including a unique username and 8-digit PIN so that they could log back in if disconnected unexpectedly or if needed to step away. Data Frame hosted as a google sheet automatically saves login info and evaluation input. Additionally, it saves randomized list of patients, and the last patient evaluated for each user account to check if evaluation is completed. Unassisted Phase page presents patient's relevant info, treatment plans including 3D dose distribution and structure, cumulative DVH, pre mid treatment 3D PET scans for NSCLC, pre and mid treatment liver functions along with pre-treatment 3D MRI scans for HCC; and contains input prompts for dose decision, decision confidence, and a textbox for remark. AI-Assisted Phase page presents AI-recommendation, outcome estimation for a range of dose show in TCP vs NTCP outcome space, and Distribution plots for all the feature variables; and contains input prompts for dose decision and decision confidence, a series of multiple-choice questions to access the user trust level, and a textbox for remark. Exit Page marks the end of the evaluation.

## Agreement with AI positively correlated with AI trust level

Under a positive AI influence, we expect evaluators to follow AI's recommendation if they agree with that recommendation. In such a scenario, the level of agreement with AI-recommendation should correlate with their level of trust on that particular recommendation. To investigate such a relationship, we defined agreement as the additive inverse of the absolute difference between AI-assisted Decision and AI recommendation, i.e. $-|aia - ai|$. Based on this definition, the level of agreement peaks at 0 when $aia = ai$, and decreases when the difference between $aia$ and $ai$ increases in either direction. As expected, we found a positive correlation between evaluators' self-reported AI trust level and their agreement with AI, as shown in Fig. 3d–f. For NSCLC, $\rho = 0.59(p < 0.001)$, for NSCLC when excluding evaluators with zero decision adjustment, $\rho = 0.69(p < 0.001)$, and for HCC, $\rho = 0.7(p < 0.001)$. The positive spearman correlation coefficient indicates an increasing monotonic relationship between AI trust level and agreement with AI. Note that the self-reported AI trust level comprises of 4 components as shown in the supplementary information.

## Mixed decision confidence trends between NSCLC and HCC

We investigated various relationships for decision confidence levels. First, we investigated the relationship between evaluators' self-reported AI-assisted decision confidence and AI trust level as shown in Fig. 4a–c. Under positive AI influence, we would expect a positive correlation between $aia\ conf$ and AI trust level. However, we only found a positive correlation for two cases: for NSCLC excluding evaluators with zero decision adjustment, $\rho = 0.34(p = 0.0094)$, and for HCC, $\rho = 0.47(p < 0.001)$. For NSCLC, we found a much weaker correlation with no significance: $\rho = 0.14(p = 0.26)$, which must be due to the two evaluators with zero decision adjustment. Then, to examine if a higher AI trust level corresponds to an increase in decision confidence, we investigated the relationship between the change in confidence level ($aia\ conf - un\ conf$) and AI-trust level as shown in Fig. 4d–f. We only found a statistically significant positive correlation for HCC i.e. $\rho = 0.32(p = 0.0055)$. For NSCLC, $\rho = 0.11(p = 0.37)$, and for NSCLC excluding evaluators with zero decision adjustment, $\rho = 0.19(p = 0.17)$. In addition, we investigated the relationship between change in confidence level and decision adjustment level, and interestingly, found opposite trends between NSCLC and HCC as shown in Fig. 4g–i. For NSCLC, $\rho = -0.24(p = 0.045)$, for NSCLC excluding evaluators with zero decision adjustment, $\rho = -0.29(p = 0.028)$, whereas, for HCC, $\rho = 0.28(p = 0.017)$. This indicates that, overall, the evaluators were more confident in reducing dose fractionation amount ($aia<un$) for NSCLC patients and increasing dose fractionation amount ($aia>un$) for HCC patients. Such behavior must have originated from the difference between the two diseases and treatment modalities as seen below.

## Higher decision confidence for the standard of care

We expect a higher confidence level for decisions close to the Standard of Care (SOC) Dose (NSCLC: 2 Gy/fx; HCC: 10 Gy/fx, fixed throughout the treatment period). As such, we investigated the relationship between decision confidence and the closeness of their decision to SOC. For this purpose, first we defined closeness as the additive inverse of the absolute difference between decision and SOC, i.e. $-|d - SOC|$, where $d$ is $un$ or $aia$. Based on this definition, closeness peaks at 0 when $d = SOC$ and decreases when the difference between decision and SOC increases in either direction. We found a positive correlation between $un\ conf$ and closeness to SOC as shown in Fig. 5a–c. For NSCLC, $\rho = 0.15(p = 0.21)$, for NSCLC excluding evaluators with zero AI-influence, $\rho = 0.25(p = 0.059)$, and for HCC, $\rho = 0.32(p = 0.0056)$. Similarly, we found a positive correlation between $aia\ conf$ and closeness to SOC as shown in Fig. 5d–f. For NSCLC, $\rho = 0.092(p = 0.44)$, for NSCLC excluding evaluators with zero decision adjustment, $\rho = 0.29(p = 0.029)$, and for HCC, $\rho = 0.48(p < 0.001)$. For NSCLC the p-value was not significant to reject the null, i.e. $\rho = 0$ (Fig. 5a, d), but as expected, once we excluded evaluators with zero decision adjustments, we found a higher correlation value with increased significance. (Fig. 5b, e).

## Inter-evaluator agreement increased with AI-assistance

We performed a concordance analysis on the decisions with intraclass correlation coefficient (ICC)[42–44] as listed in Fig. 6 and Table 1. We compared four types of ICCs between unassisted and AI-assisted decision for both NSCLC cohort and HCC cohort as shown in Fig. 6a, b, respectively. We found that on average the ICC between evaluators increased with AI-assistance for both cohorts. The higher ICC for AI-assisted decisions indicates that AI-assistance resulted in decrease in inter-evaluator (inter-physicians) variability by reducing the uncertainty in decision-making.

## Decisions adjusted for different endpoints among NSCLC and HCC cases

In the AI-assisted Phase, along with AI recommendation, evaluators were also provided with RT outcome estimates (RTOE). So, we analyzed decision adjustment behavior with respect to outcome estimates as shown in Fig. 7 for NSCLC and Fig. 8 for HCC. Outcome estimates were provided in a two-dimensional outcome space spanned by TCP and NTCP. Figures 7a and 8a summarize outcome estimates for unassisted decision and Figs. 7b and 8b for AI-assisted decision, grouped by individual patient, and color-coded and marked by individual evaluator.

To investigate the effect of outcome estimates on decision-making, we isolated the adjusted decisions ($un \neq aia$) and analyzed the frequency of increase/decrease in TCP (Figs. 7c and 8c) and NTCP (Figs. 7d and 8d). For NSCLC, we observed that 76% (31/41) of decision adjustments resulted in increase in TCP and NTCP and for HCC, we observed that 74% (25/34) of decision adjustments resulted in decrease in TCP and NTCP. Based on the rationale that the clinical goal of RT is to increase TCP while decreasing NTCP, we deduced that the majority (76%) of NSCLC decisions were adjusted to increase TCP while the majority (74%) of HCC decisions were adjusted to decrease the NTCP.

In the absence of ground truth, we analyzed the adjusted decision based on the estimated outcome and a derived scoring schema of toxicity free local control, i.e., $TCP(1 - NTCP)$. The monotonic schema has a maximum value of 1 for ideal clinical outcome of $(tcp, ntcp) = (1, 0)$ and a minimum value of 0 for the dose-limiting factor $ntcp = 1$. The higher value indicates a higher TCP and a lower NTCP. Figures 7e and 8e summarize the change in scores between

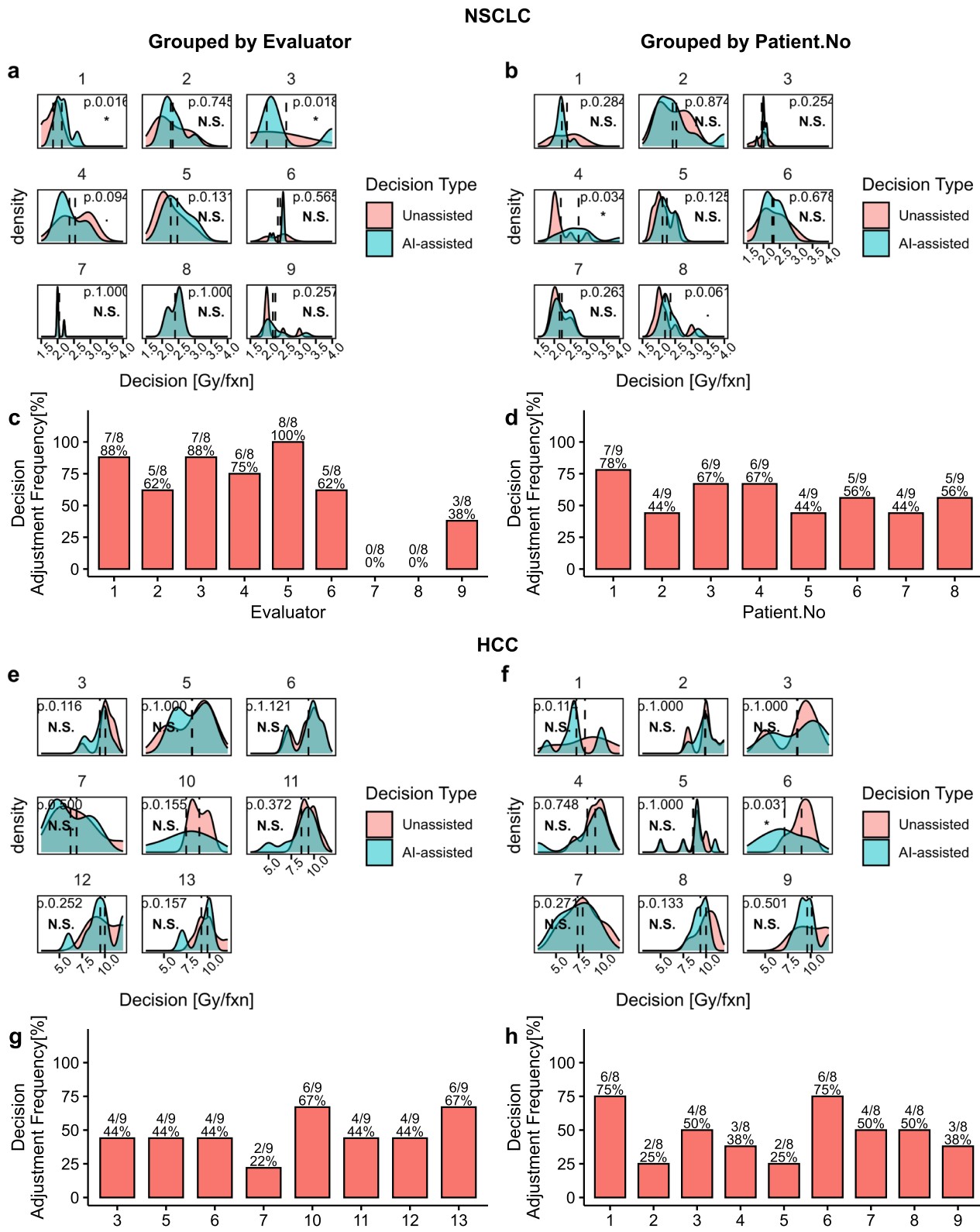

**Fig. 2 | Unassisted vs AI-assisted decision analysis grouped by evaluators and patient number.** Plots **a**–**d** summarizes data from NSCLC and **plots e**–**h** from HCC. **Density plots a**, **b**, **e**, and **f** compare the unassisted (*un*) and AI-assisted (*aia*) distribution and includes the p-values from a matched pair randomization two-sided t-test with *un* − *aia*≠0 as the alternative hypothesis and standard significance code: *** (*p*<0.001), ** (*p*<0.01), *(*p*<0.05),.(*p*<0.1), N.S (*p* ≥ 0.1). Bar plots **c**, **d**, **g**, and **h** shows the frequency of decision adjustment (*un*≠*aia*) after AI-assistance in percentages.

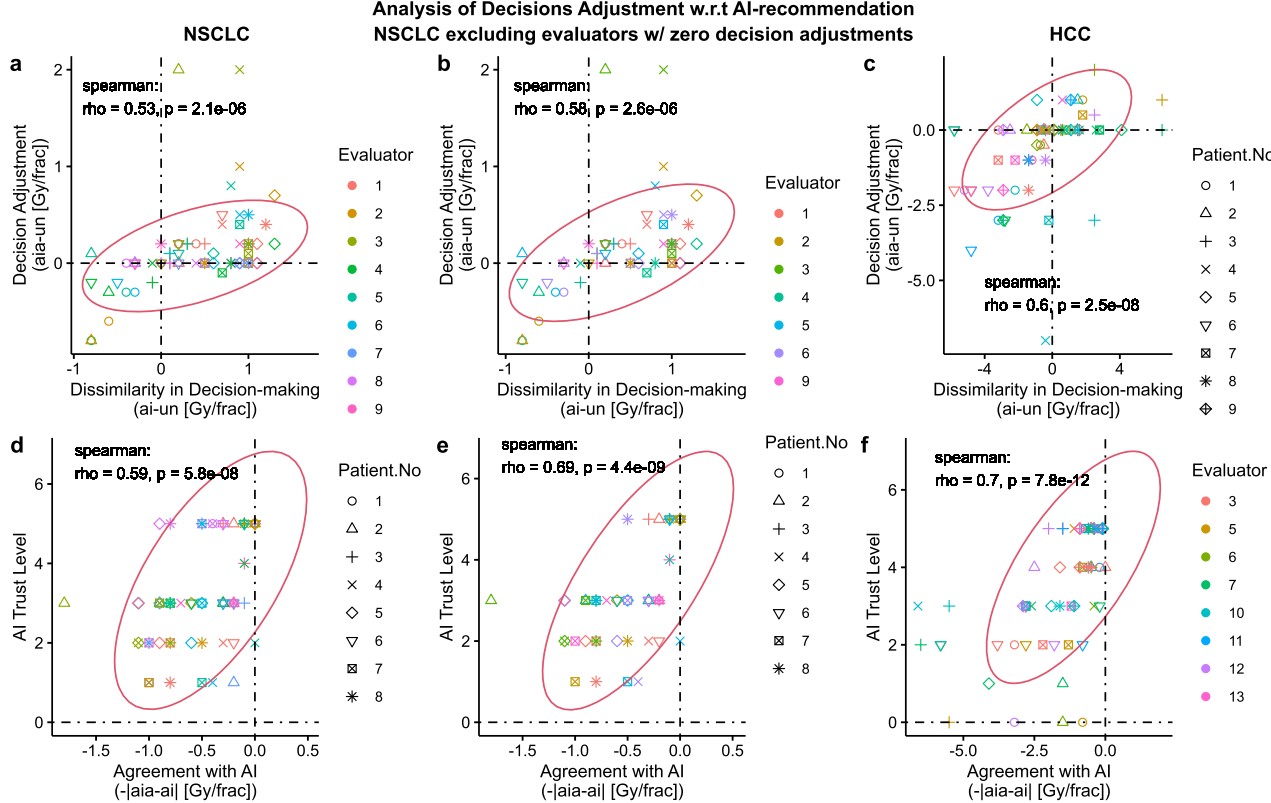

**Fig. 3 | Analysis of Decision adjustment with respect to AI recommendation.**
The first, second, and third column of figures correspond to NSCLC, NSCLC excluding evaluators with zero decision adjustment, and HCC, respectively, and all the plots are colored by evaluators and marked coded patient number. All evaluators in HCC adjusted at least one of their decisions. **Scatter plots a, b,** and **c** shows relationship between the level of decision adjustment ($aia - un$) and dissimilarity with AI recommendation ($ai - un$). **Scatter plots d–f** show relationship between AI Trust Level (0-5, 5 being the highest level) and agreement with AI-recommendation ($-|aia - ai|$). The $aia - ai = 0$ line corresponds to absolute agreement with the AI-recommendation. All plots include the Spearman correlation coefficients, p-values (two-sided t test), and co-variance ellipse (95 % confidence). Covariance ellipses are included for a visual insight about the data distribution.

adjusted unassisted and AI-assisted decision, grouped by patient. We didn't observe a clear trend. Then to get sense of the overall trend, we calculated pairwise difference in the scoring (*aia score−un score*) and summary statistics as shown in Figs. 7f and 8f, respectively. For NSCLC, we found a conflicting mean and median, where the mean showed an increase in scoring (desirable) while the median showed a decrease. For HCC, however, both mean and median showed improvement in decisions. Nevertheless, in both cases, we found a right-skewed distribution which indicates an overall improvement in decisions due to AI-assistance.

## Analysis of evaluators' remarks
As a final step, we analyzed evaluators' self-reported text remarks as listed in supplementary information. First, we summarized the remarks and then selected and counted the keywords. As the remarks were optional, we received only 48 total remarks: 14 from the NSCLC Unassisted Phase, 14 from the NSCLC AI-assisted Phase, 9 from the HCC Unassisted Phase, and 11 from the HCC AI-assisted Phase.

For NSCLC Unassisted Phase, we found that controlling the radiation-induced toxicity for organs at risk, especially for the esophagus, was the main factor for making dose escalation or de-escalation decision, which was mentioned 9 times, followed by patients' KBR-ART evaluation phase dose-response (3 times). Other remarks pertained to disagreement with KBR-ART evaluation phase dose fractionation (1 time), disagreement with dose-volume histogram curve (2 times), preference for anatomical adaptation instead of dose adaptation (1 time), and trouble with PET treatment planner viewer

(1 time). We note that anatomical adaptation is a complementary adaptive strategy where a treatment plan is adapted to physiological changes such as weight loss or evolving tumor shape and size. For HCC Unassisted Phase, the summary of the remark indicated that the main decision-making scheme was to escalate/de-escalate dose over a certain amount of biologically effective dose (6 times) based on the KBR-ART evaluation phase response (4 times) in terms of liver function (4 times) and Child Pugh Score (3 times). Other remarks pertained to organs at risk (1 time) and large tumor size (1 time).

For NSCLC AI-assisted Phase, we found that the RT outcome estimation curve (2 times), especially the estimation for NTCP (5 times) played an important role in the decision adjustment. The main concern pertained to the lack of steep change in NTCP for a large change in the dose fractionation value and small error bars. There were concerns over the relation between RT outcome estimation and AI recommendation (4 times) and organs at risk toxicity (4 times). Other remarks pertained to large tumor volume (1 time) and whether chemotherapy information was included (1 time). The remark from evaluator #7 (one of the two evaluators with zero decision adjustment) stood out as it stated that prior clinical trials showed no benefits of dose escalation providing insight into their collaborative decision-making process. For HCC AI-assisted Phase, we again found that RT outcome estimation (6 times) played an important role in decision-making. In contrast to NSCLC, one of the concerns was the steep rise in NTCP for increasing dose fractionation. We also found that evaluators directly agreed/disagreed with the AI (4 times). Other remarks pertained to organs at risk (1 time) and biologically effective dose (1 time).

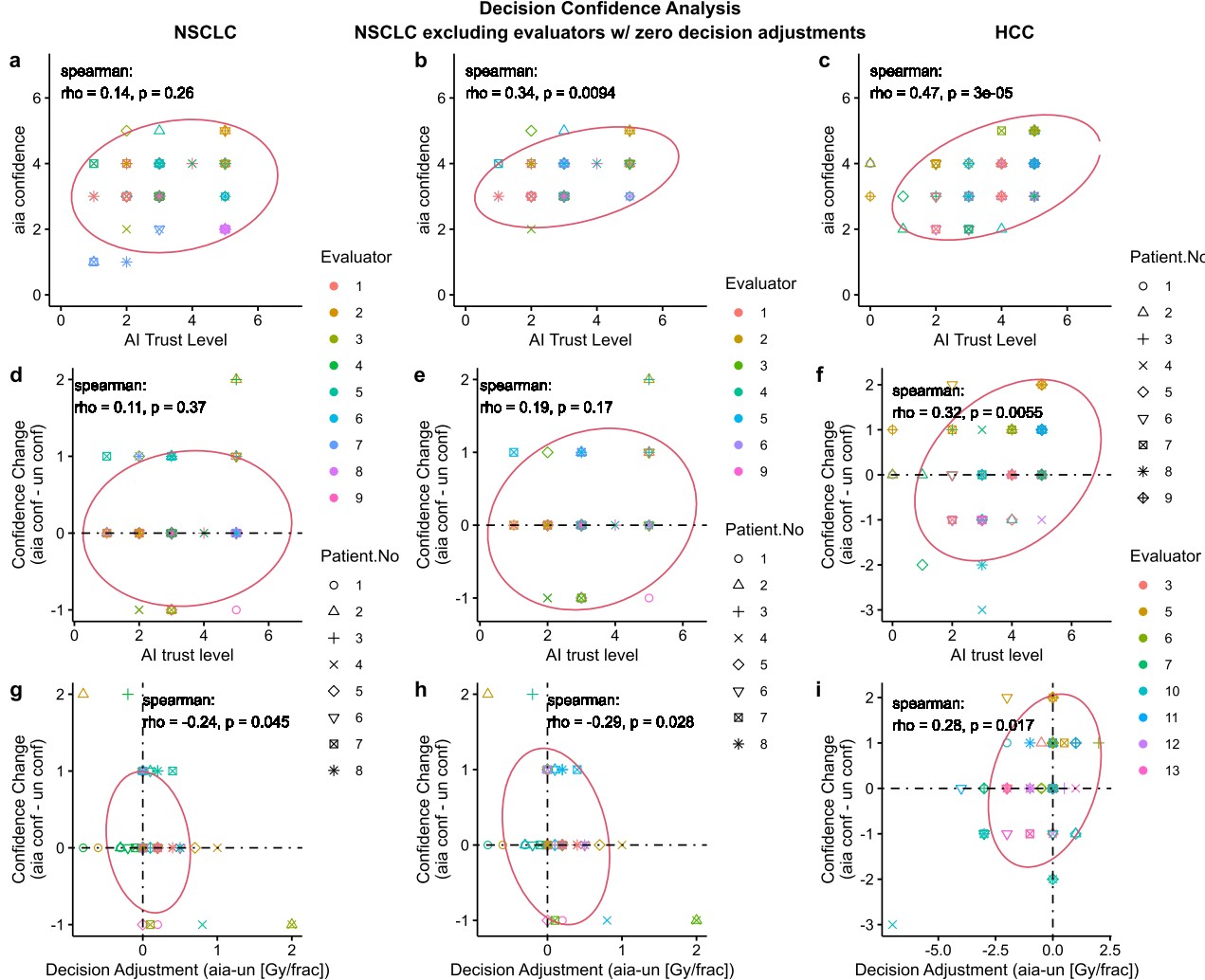

**Fig. 4 | Analysis of evaluator's decision confidence.** The first, second, and third column of figures correspond to NSCLC, NSCLC excluding Evaluators with zero decision adjustment, and HCC, respectively. All evaluators in HCC adjusted at least one of their decisions. **2D Scatter plots a, b**, and **c** show the relationship between evaluators' AI-assisted decision (*aia*) confidence (0-5, 5 being the highest level) and *AI Trust level* (0-5, 5 being the highest level). **2D Scatter plots d, e**, and **f** show the relationship between evaluators' change in confidence level (*aia conf − un conf*) and *AI trust level*. **2D Scatter plots g, h** and **i** show the relationship between evaluators' change in confidence level and level of decision adjustment with AI-assistance. All plots include the Spearman correlation coefficients, p-values (two-sided t test), and co-variance ellipse (95 % confidence). Covariance ellipses are included for a visual insight about the data distribution.

## Discussion

ARCliDS is designed for therapeutics and is fundamentally different from diagnostic AI platforms which are trained with known labels with a supervised learning framework. In KBR-ART, retrospective clinical endpoints (local control and toxicity) are available for only one decision point and thus lack ground truth for validation in general; and treatment options (dose/fraction) are continuous variables (NSCLC: 1.5-4 Gy/fx; HCC: 1-15 Gy/fx) and fundamentally differ from categorical decisions. As such, our study is unique in comparison to other Human−AI interaction studies such as collaborative decision-making with image-based visual diagnostic AI in skin cancer[45], with diagnostic AI for lesion detection and categorization from colonoscopy[15], with diagnostic AI for malignant nodules detection from chest radiographs[46], and with image-based AI for chemotherapy response assessment in bladder cancer from CT urography[47]; in which either decision option was categorical (e.g. multi-class image classification), or ground truth was available (e.g. labeled image or retrospective post-treatment tumor stage) or both. A similar reinforcement learning (RL) framework is presented by Barata et al.'s[48] interaction study of improved diagnostic decision-making in skin cancer with expert-

generated reward functions indicating the value of RL for medical decision making[49]. Moreover, whereas categorical decision with the availability of ground truth simplifies evaluation, continuous decisions provide opportunities for conducting high-resolution correlation analysis. Furthermore, dose options have a natural ordering property with respect to the radiobiological principle, i.e., higher dose corresponds to higher TCP as well as higher NTCP compared to lower doses. Thus, in this study along with analyzing the observed variables−decisions, decision confidences, and AI trust level−we were able to derive a number of quantities to investigate human−AI interaction and collaborative decision-making behavior. In particular, we investigated decision adjustment frequency, decision adjustment level, dissimilarity in decision-making with AI, agreement with AI recommendation, and closeness of decision with standard of care. We focused our analysis on model-agnostic and model-specific behavioral changes due to AI-assistance.

The combination of the following two levels of seemingly conflicting observations suggests that AI-assistance does not uniformly influence an expert's decision-making process: (i) statistically non-significant difference between unassisted and AI-assisted decisions,

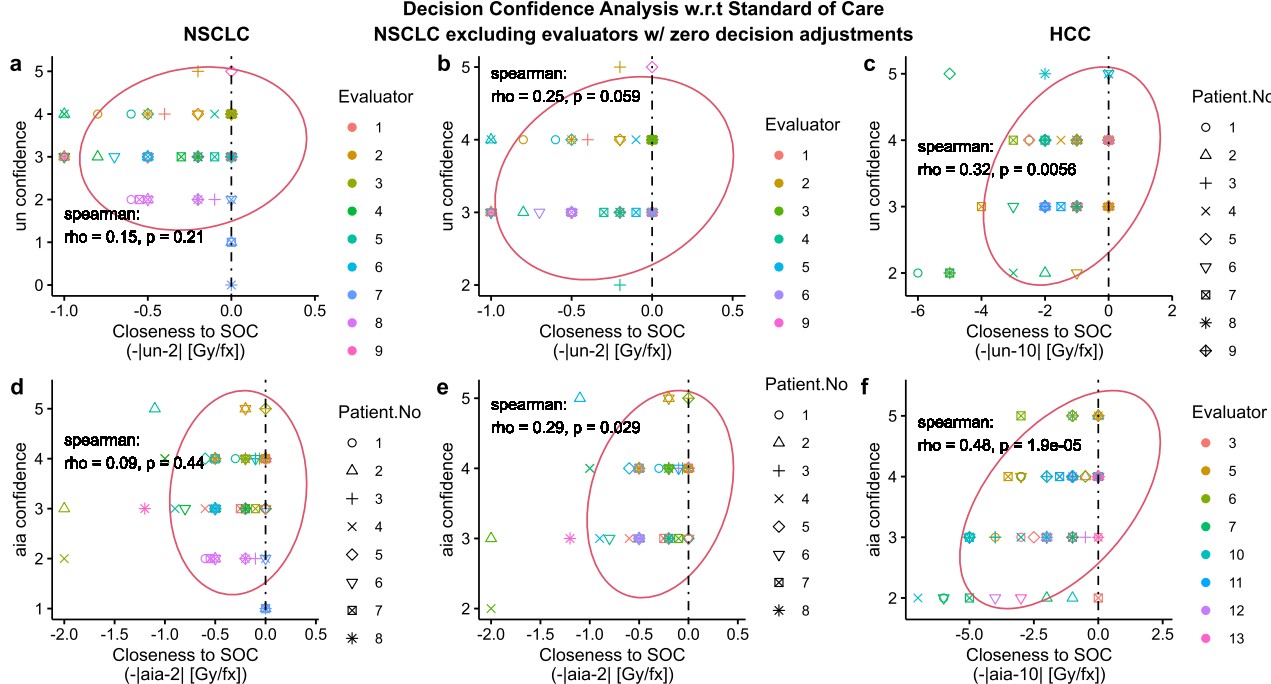

**Fig. 5 | Analysis of Evaluator's decision confidence with respect to the Standard of Care Dose fractionation (2 Gy/fx for NSCLC RT; 10 Gy/fx for HCC SBRT).** The first, second, and third column of figures correspond to NSCLC, NSCLC excluding evaluators with zero decision adjustment, and HCC, respectively. All evaluators in HCC adjusted at least one of their decisions. 2D Scatter plots **a**–**c** show the relationship between unassisted decision (*un*) confidence (0-5, 5 being the highest level) and closeness of *un* to the standard of care dose decision values

(NSCLC:$-|un - 2|$Gy/fx); HCC: $-|un - 10|$Gy/fx). 2D Scatter plots **d**, **e**, and **f** show the relationship between AI-assisted decision (*aia*) confidence and closeness of *un* to the standard of care dose decision values (NSCLC:$-|aia - 2|$ Gy/fx); HCC: $-|aia - 10|$Gy/fx). All plots include the Spearman correlation coefficients, p-values (two-sided t test), and co-variance ellipse (95% confidence). Covariance ellipses are included for a visual insight about the data distribution.

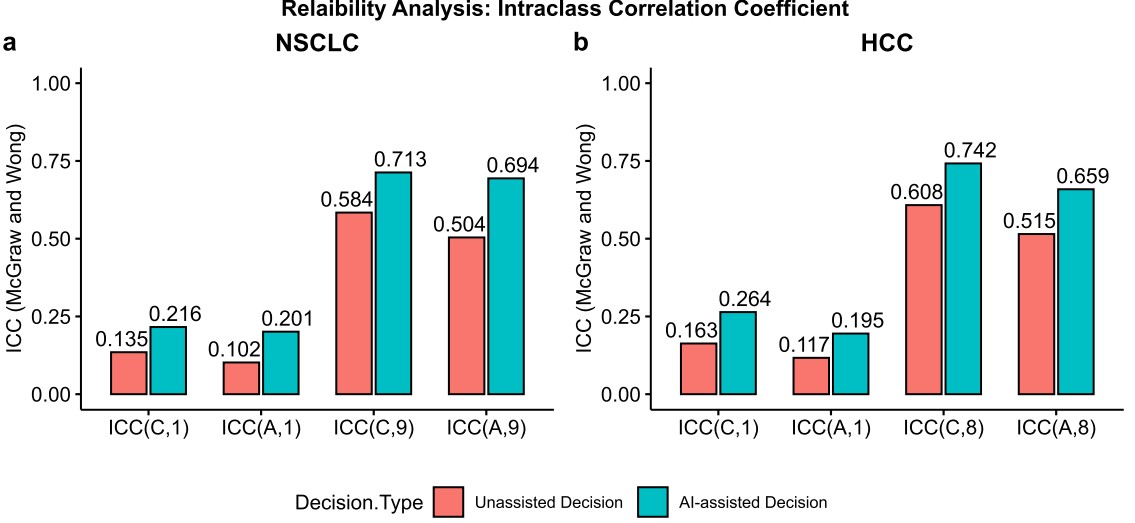

**Fig. 6 | Reliability analysis via intraclass correlation coefficient (ICC). Bar plots a** and **b** compare McGraw and Wong's ICC between unassisted decision and AI-assisted decision for NSCLC and HCC, respectively. ICC value along with 95% confidence interval, and p-value (one-sided F-test, $H_\alpha : icc > 0$) for NSCLC and HCC are presented in Table 1. We applied two-way random effects model to calculate four types of ICC for $n \times k$ data structure where $n$ and $k$ are the number of patients and evaluators, respectively, which were both chosen randomly from a larger pool

of patients and evaluators (NSCLC: $n = 8$, $k = 9$; HCC: $n = 9$, $k = 8$). ICC type Consistency (C) measures the symmetric differences between the decisions of the $k$ evaluators, whereas ICC type Absolute Agreement (A) measures the absolute differences. ICC unit Single rater corresponds to using the decision from a single evaluator as the basis for measurement and ICC unit Average corresponds to using the average decision from all evaluators.

yet considerable decision adjustment frequency and (ii) absence of decision adjustment from evaluators #7 and #8 for NSCLC, yet two decision adjustments from evaluator #7 for HCC. In particular, upon further investigation of evaluator #7's remark, we found that evaluator

#7 may have a preference against dose escalation in NSCLC but is open to adaptation in HCC. The above conclusion comes from comparing the following two remarks: (i) in the AI-assisted Phase of the NSCLC module, evaluator #7 left a remark for patient 8, "*with multiple level 1*

**Table 1 | Intraclass correlation coefficient (McGraw and Wong)**

| Disease | ICC Type | Unassisted Decision | AI-assisted Decision |
|---------|----------|---------------------|----------------------|
| NSCLC | ICC(C,1) | 0.135 [−0.005, 0.507] (p = 0.032) | 0.216 [0.04, 0.606] (p = 0.004) |
| NSCLC | ICC(A,1) | 0.102 [−0.003, 0.426] (p = 0.031) | 0.201 [0.039, 0.584] (p = 0.003) |
| NSCLC | ICC(C,9) | 0.584 [−0.049, 0.902] (p = 0.032) | 0.713 [0.275, 0.933] (p = 0.004) |
| NSCLC | ICC(A,9) | 0.504 [−0.053,0.871] (p = 0.035) | 0.694 [0.264, 0.927] (p = 0.003) |
| HCC | ICC(C,1) | 0.163 [0.006, 0.521] (p = 0.019) | 0.264 [0.069, 0.631] (p = 0.001) |
| HCC | ICC(A,1) | 0.117 [0.004, 0.427] (p = 0.019) | 0.195 [0.045, 0.537] (p = 0.001) |
| HCC | ICC(C,9) | 0.608 [0.049, 0.897] (p = 0.019) | 0.742 [0.373, 0.932] (p = 0.001) |
| HCC | ICC(A,8) | 0.515 [0.003, 0.858] (p = 0.024) | 0.659 [0.254, 0.904] (p = 0.002) |

*trials showing no benefit of dose escalation, it's hard to have any confidence in these recommendations*"; (ii) in the AI-assisted Phase of HCC module, evaluator #7 left a remark for patient 6, "*with TCP near identical per dose, it seems like omitting last two fractions should be an option*" and adjusted their $un = 7$ Gy/fx to $aia = 4$ Gy/fx, seemingly agreeing with $ai = 4.2$ Gy/fx. Thus, we deduce that collaborative decision-making is a highly dependent process which in our case depended on the clinician's prior knowledge, the patient's condition, disease type, and treatment modality.

The observation of a positive correlation between decision adjustment level ($aia − un$) and dissimilarity in decision-making ($ai − un$); and between AI-trust level and agreement with AI ($−|aia−ai|$) makes it clear that understanding AI-influence in collaborative decision-making needs beyond binary considerations. The simplest of such considerations involves two conditions: first, whether the clinical experts believe in the benefits of AI system, and secondly, if they do believe in the AI, the level of agreement with AI's recommendations and outcome prediction. In a discretized scenario, there can be four possibilities as shown in Table 2.

Clinicians that do not believe in an AI system (e.g. evaluators #7 and #8 for ARCliDS-NSCLC), are generally not influenced by any AI recommendations and tend not to adjust their decision ($un = aia$). However, even when a clinician believes in the AI system, if their unassisted decision happens to be the same or close to AI recommendation ($un = ai$), then they will again not adjust their decision ($un = aia$), and we wouldn't be able to determine AI-influence from such cases. Conversely, when clinicians make decision adjustments ($un ≠ aia$) then we can consider AI-influence regardless of their agreement/disagreement with the AI recommendation. However, when evaluators adjust their decision even when their $un = ai$, then we can conclude that they must have made adjustments based on the outcome prediction (TCP/NTCP) bypassing AI recommendation. The summary of text remark presented in supplementary information presents examples of such cases where evaluators adjusted decisions based on AI-recommendation and also based on RT outcome estimation by passing AI-recommendation. Note that here we make a distinction between belief as overall trust and trust level as a trust on individual recommendation, because we observed that, although, both evaluators #7 and #8 didn't change any decisions, they reported nonzero AI trust level for all recommendations. The minimum, median, and maximum AI trust levels reported by evaluator #7 were 1, 3, and 5 respectively, whereas evaluator #8 reported an AI trust level of 5 for all patients.

The model-specific behavioral changes were observed in various analysis. In both NSCLC and HCC, we observed a significant positive correlation between *aia conf* and AI trust level indicating clinical experts were more confident in their decision when they trusted a particular AI-recommendation. However, only the HCC cohort exhibited a significant positive correlation between change in confidence level (*aia conf* − *un conf*) and AI-trust level indicating that only HCC experts' confidence grew with AI-assistance. Similarly, for both cases,

we found a positive correlation between decision confidence levels (*un conf* and *aia conf*) and closeness of decisions to the standard of care (SOC) which indicates that evaluators were generally more confident in prescribing dose fractionation closer to the clinical practice. While this is a reasonable behavior, the correlation value and significance level for NSCLC were much lower than that of HCC. Moreover, we observed a higher correlation between *aia conf* and closeness to SOC, which may be due to evaluators feeling validated by the AI. When AI recommends doses closer to SOC, we would expect an amplification of confidence level due to the compounding effect of the evaluators being comfortable in repeating day-to-day practice and AI's confirmation.

Model-specific behavior changes can be attributed to the difference in treatment options, organ and cancer type, and treatment protocols. NSCLC patients were administered RT in 30 fractions at the rate of 2 Gy/fx while HCC patients were administered SBRT in 5 fractions at the rate of 10 Gy/frac. The higher rate of dose causes more concern for the toxicities. Another difference is the event rate of the cohort: 95 out of 99 HCC patients showed local control (Table S10 in Niraula et al.[22]). which is much higher than in NSCLC. Such underlying distribution is learned by AI outcome prediction model for HCC which made a high TCP projection for all patients and only changed slightly for the full range of dose decision values. Note that such behavior is similar to a type of AI bias shown by language models where AI learns slightly differently for underrepresented samples[13]. Additionally, for NSCLC pneumonitis was the main toxicity of concern, and for HCC liver function, which was measured in Child-Pugh score. These differences may have led to different decision adjustment behaviors, in particular increasing in TCP for NSCLC and decreasing NTCP for HCC.

The user-reported remarks provided miscellaneous feedback providing us with additional insights. We found that the organs at risk and the patient's treatment response were the top priorities, consistent with the clinical practice where toxicity is the dose-limiting factor. For NSCLC, evaluators were concerned about esophagus and lung toxicity, and for HCC, liver function and heart toxicities. We found that evaluators carefully inspected AI's outcome prediction for not only the AI-recommendation but also for the whole range of decisions. In particular, evaluators closely monitored the NTCP and sometimes bypassed AI's recommendation to lower the complication probability. This is a clinically desired behavior, as over-reliance on AI recommendations could lead to erroneous decisions. However, we note that since we received an unequal number of remarks from a handful of evaluators, the summary will be swayed toward the evaluators with the higher number of remarks.

To recap, Human−AI interaction in dynamic decision-making has high variability as it depends on complex interrelationship between the expert's prior knowledge and preferences, the patient's state, disease site, treatment modality, and AI behavior. The collaborative decision-making for treating advanced diseases can be summarized as follows: (i) clinicians may not believe in an AI system, completely disregarding AI's recommendation (ii) clinicians may believe in the AI

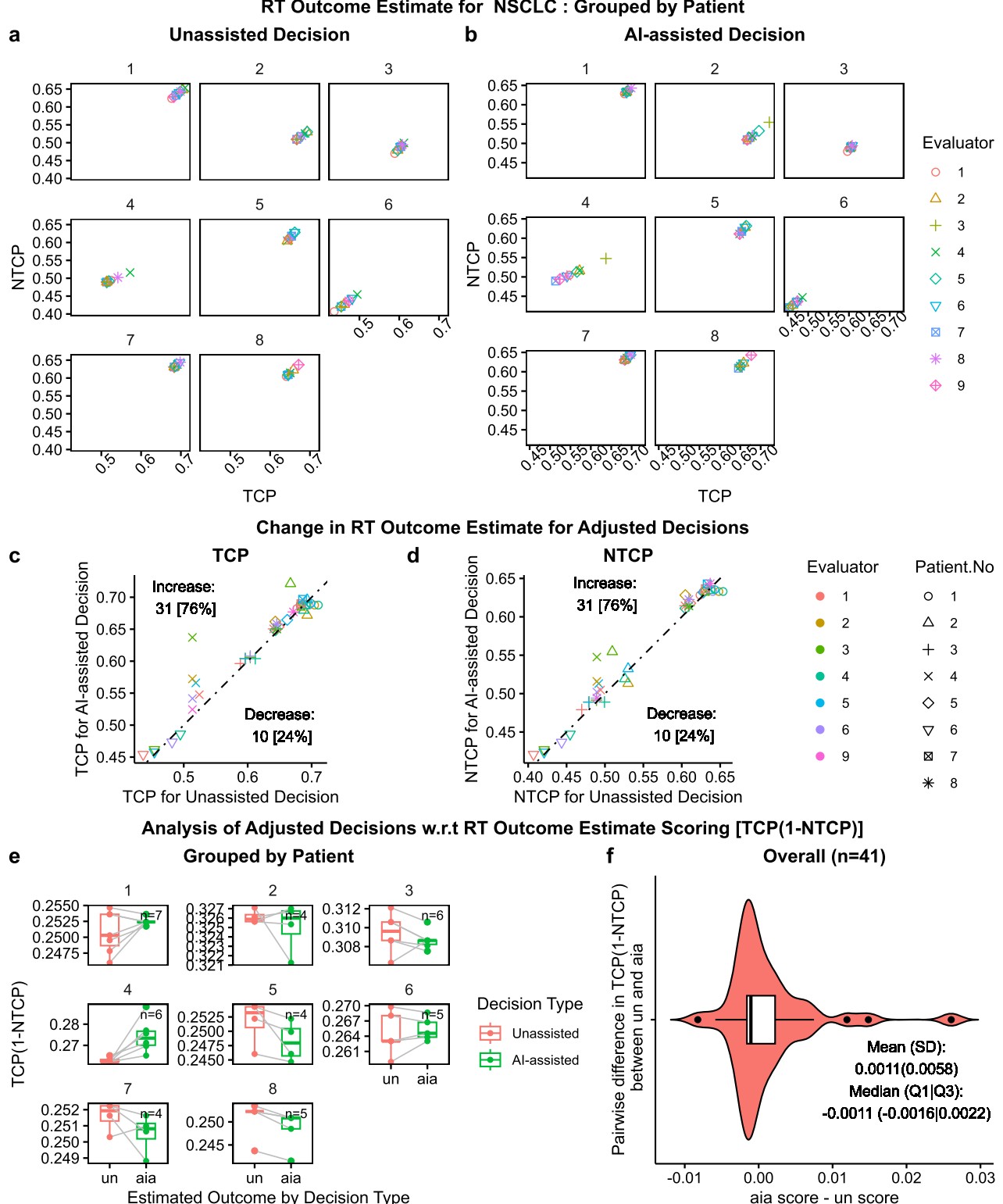

**Fig. 7 | AI utility analysis of decision adjustment with respect to RT outcome estimate for NSCLC.** Scatter plots **a** and **b**, grouped by patient number, show the RT outcome estimate (RTOE) in the space spanned by tumor control probability (*TCP*) and normal tissue complication probability (*NTCP*) for Unassisted (*un*) and AI-assisted (*aia*) decision, respectively. Scatter plots **c** and **d** show the change in RTOE for adjusted decisions in *un* vs *aia* *TCP* space and *un* vs *aia* *NTCP* space, respectively, including the 45° Null dashed line. Out of 41 decision adjustment, 32 (76%) increased both *TCP* and *NTCP* while 10 (24%) decreased *TCP* and *NTCP*.

Paired plot e and violin plot f, present analysis of adjusted decision based on RTOE scoring schema $TCP(1 - NTCP)$ [1 for $(tcp, ntcp) = (1, 0)$, 0 for $ntcp = 1$]. Paired plots **e** compares the change in score for *un* and *aia* for each patient. Violin plot **f** presents the overall summary statistics for the pairwise difference in score between *aia* and *un*: $mean(sd) = 0.0011(0.0058)$; $median(Q1|Q3) = -0.0011(-0.0016|0.0022)$. Box plots include center line: median, box limits: upper and lower quartiles; whiskers: 1.5x interquartile range; and points: outliers.

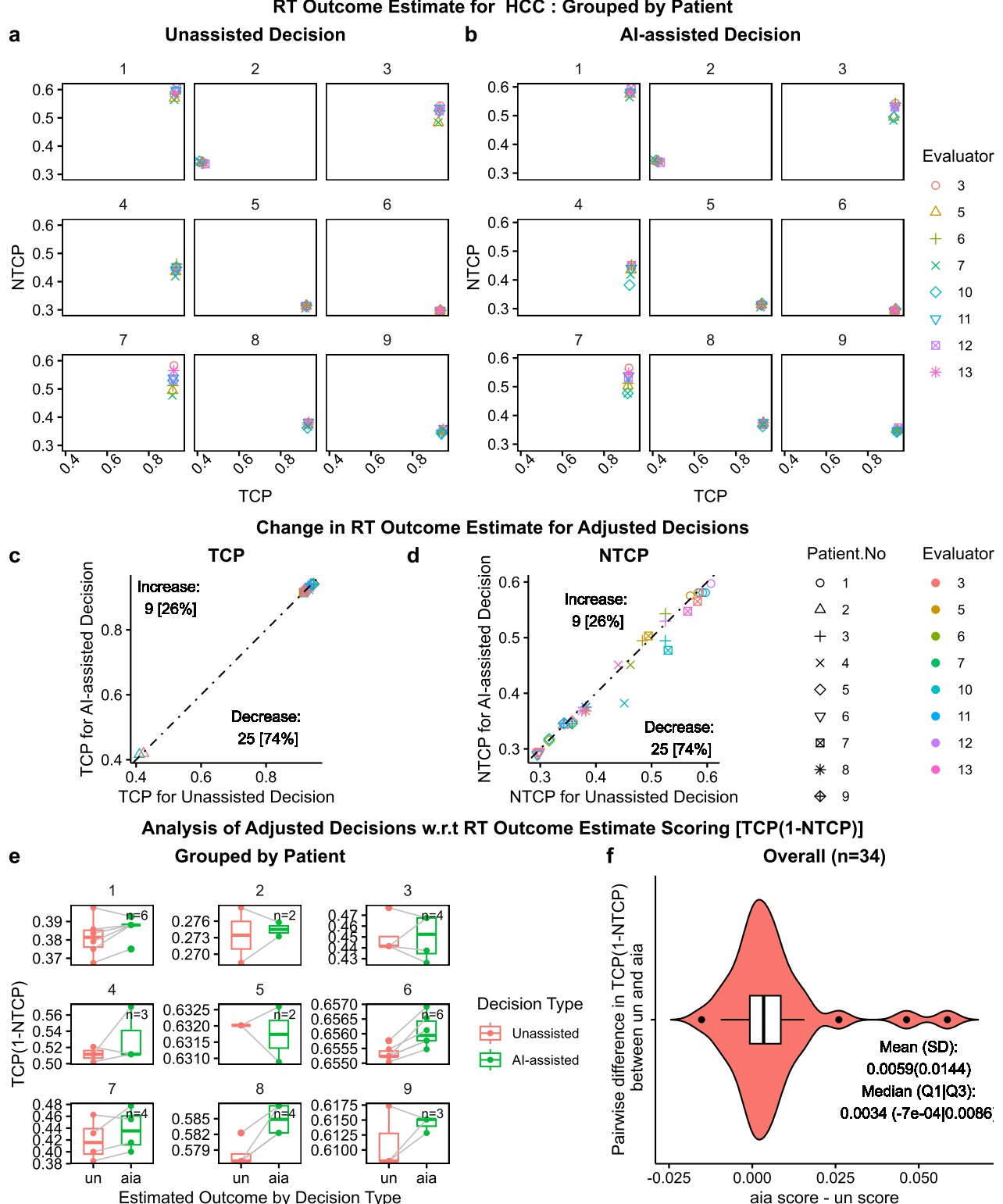

**Fig. 8 | AI utility analysis of decision adjustment with respect to RT outcome estimate for HCC.** Scatter plots **a** and **b**, grouped by patient number, show the RT Outcome Estimate (RTOE) in the space spanned by tumor control probability (*TCP*) and normal tissue complication probability (*NTCP*) for Unassisted (*un*) and AI-assisted (*aia*) decision, respectively. Scatter plots **c** and **d** show the change in RTOE for adjusted decisions in *un* vs *aia TCP* space and *un* vs *aia NTCP* space, respectively, including the 45° Null dashed line. Out of 34 decision adjustment, 9 (26%) increased both *TCP* and *NTCP* while 25 (74%) decreased *TCP* and *NTCP*. Paired plots **e** and violin plot **f**, present analysis of adjusted decision based on RTOE scoring schema $TCP(1 - NTCP)$ [1 for $(tcp, ntcp) = (1, 0)$, 0 for $ntcp = 1$]. Paired plots **e** compares the change in score for *un* and *aia* for each patient. Violin plot **f** presents the overall summary statistics for the pairwise difference in score between *aia* and *un*: $mean(sd) = 0.0059(0.0144)$; $median(Q1|Q3) = 0.0034(-7E - 4|0.0086)$. Box plots include center line: median, box limits: upper and lower quartiles; whiskers: 1.5× interquartile range; and points: outliers.

**Table 2 | AI influence truth table**

|  | No decision adjustment after AI-assistance ($un = aia$) | Decision Adjustment after AI-assistance ($un \neq aia$) |
|---|---|---|
| Similar Human-AI decision ($un = ai$) | Not Sure | Yes (based on RTOE) |
| Dissimilar Human-AI decision ($un \neq ai$) | No | Yes |

system but will critically analyze AI recommendations on a case-by-case basis, (iii) When clinicians find AI recommendations beneficial to patients they will adjust their decision as necessary, (iv) When clinicians do not find AI recommendations beneficial they will either stick to their own decision or, if an outcome prediction model is available, will search for an optimal decision on their own bypassing AI-recommendation. AI-assistance can reduce inter-physician variability and improving model transparency and explainability helps in balancing the reliance on AI. Clinicians are generally more comfortable making decisions that align with the standard clinical practice but will prescribe non-standard treatment if deemed optimal and necessary especially to lower treatment side effects. Such model-agnostic behavior should be true for other therapeutic medical domains.

There are a few limitations to our study as discussed next. Adaptive treatment strategies are emerging and yet to be widely incorporated in standard clinical practice. Besides KBR-ART, there exist several adaptive strategies, for instance, adaptation of the number of dose fractionation with fixed dose per fraction[22,28], during treatment anatomical adaptation via CT-guided[50] or MRI-guided strategies[51], and gene expression-based adaptation where the full course of radiotherapy is personalized[52]. Thus, evaluators' level of trust and confidence on AI recommendation would have been higher and more representative if the AI had been designed for standard clinical practice. Nevertheless, the study methodology provides a direction for future collaborative decision-making study on AI for standard adaptive strategies.

A limitation of the emerging R-ART field is the general unavailability of representative external validation datasets. In this study, patients were randomly selected from the original dataset and thus we expect the AI predictions to be generally better than those on an external dataset but within the estimated confidence levels. However, the primary purpose of this study is to evaluate AI's influence on physicians' clinical decisions rather than to assess AI's performance. Inclusion of patients from the original dataset would not have caused data/information leakage into the study because the evaluators had never seen the patients before and the retrospective outcome and clinical decision were withheld from them. Additionally, we didn't find any evidence of overreliance on AI, however, we acknowledge that future studies should include an external validation set to minimize potential bias.

Although this study generated 144 evaluations because the data is two-dimensional and from two studies, the result from our analysis will have low inferential capability. An obvious way is to increase the sample size. However, because of the two-dimensional nature of the data, increasing sample size increases the effort quadratically ($\sim n^2$). To complicate the matter, we found that the collaborative decision-making process depends on many factors and is sensitive to situational change. A similar observation was reported in a Human–AI interaction study by McIntosh et al.[53], where medical professionals were found to behave more conservatively during prospective study than in retrospective settings. We expect such risk-associated behavior-shift in a prospective setting. As such, to infer clinical collaborative decision-making behavior, conducting a prospective study should take priority followed by ensuring a sufficient sample size. Additionally, a randomized prospective study would also be necessary to validate AI's clinical utility[54].

We did not analyze decision-making behaviors of sub-groups of evaluators stratified by experience, specialty, and affiliation, because

further stratifying data would push to an extreme the sample size-related limitation. However, we want to note results from previous studies: Tschandl et al.[45], in their human-computer collaboration for skin cancer recognition study, found that that the least experienced clinicians gained the most from diagnostic AI support, however, faulty AI could mislead all clinicians irrespective of experience level. Reverberi et al.[15], reported similar behavior, where non-expert endoscopist (less than 500 colonoscopies performed) gained more from AI-assistance (improved accuracy) however, they found that experts were less able to discriminate between good and bad AI advice, and expert's average confidence was lower toward both their own judgments and AI recommendation. Similarly, Sun et al.[47] reported that AI-assistance helped inexperienced (resident and fellows) evaluators to increase their accuracy of assessing the bladder cancer treatment response. They found that with AI-assistance, inexperienced and experienced evaluators attained a similar accuracy. In the other hand, they found different performance among evaluators stratified by specialty: oncologists' accuracy improved more than that of radiologists. In contrast, Lee et al.[46] reported that the evaluator's characteristics, including experience level, were not associated with accurate AI-assisted readings of chest radiographs.

There are multiple avenues for extending the current study design. Similar to Tschandl et al.[45] and Gaube et al.[55] work for evaluating the effect of diagnostic AI's advice quality and intent, we could investigate whether a good or a faulty/inaccurate AI-support would affect the collaborative decision-making process differently. This is especially important to understand the risk of faulty AI and misinformation on AI-influence. Investigating clinicians' response to varying levels of AI explainability, time constraints, and simulated risk would be other clinically meaningful extensions. On the other hand, stratifying the analysis by the characteristics of clinicians, patients, and diseases could provide further understanding of the collaborative decision-making process. Such knowledge would be valuable for the personalization of future medical AI on the clinician's level according to their professional background and on the patient level according to the patient's preferences when applicable.

To balance the reliance on AI system, improving model transparency and explainability is essential for future AI model development. Inclusion of an outcome prediction model is recommended for therapeutic AI decision-support systems. Whereas personalization of treatment focuses on capturing inter-patient heterogeneity for optimal treatment outcome, personalization of AI system based on clinician's background level will further help in optimizing treatment outcome by minimizing inter-physician variability. As such, online fine-tuning of AI per user should be an option. Therapeutic AI system's learned behavior will reflect the subtle differences originating from the differences in disease type and treatment modalities. Evaluating therapeutic AI's behavior based on traditional supervised learning validation techniques will fall short due to the fundamental difference in the nature of the problems. Thus, application-grounded Human–AI interaction evaluation study involving domain-expert is essential for validating Therapeutic AI systems.

## Methods

### Knowledge based response-adaptive radiotherapy (KBR-ART)

KBR-ART[21–24] is a dynamic interventive treatment strategy, collectively known as a dynamic treatment regimen (DTR)[5,6], which consists of at least three phases: Pre-Treatment Assessment, Treatment Response

Evaluation (evaluation phase), and Treatment Adaptation (adaptation phase). The evaluation phase begins with the start of treatment and lasts up to the intervention; the adaptation phase follows and lasts for the remaining treatment period. In the pre-treatment phase, a patient's disease and condition is assessed and a treatment plan is tailored. In the evaluation phase, a patient's treatment response is evaluated by comparing pre and mid treatment multi-omics information changes. Based on the treatment responses, the patient's associated outcome probabilities are estimated. In the adaptation phase, treatment planning is adapted for a personalized and optimal outcome.

## Adaptive radiotherapy clinical decision support (ARCliDS)

ARCliDS is a web-based clinical decision support software for AI-assisted optimal decision-making in KBR-ART[22]. Given a patient's pre and during treatment multi-omics information, ARCliDS can estimate treatment response and recommend an optimal intervention for the remaining treatment period. The treatment response is estimated in terms of tumor control probability (TCP) and normal tissue complications probability (NTCP), and the intervention is recommended in terms of daily dose fractionation. The optimal intervention corresponds to the maximum TCP and minimum NTCP estimate. Additionally, ARCliDS provides uncertainty estimates via statistical ensemble for both outcome estimates and dose recommendation.

ARCliDS can be divided into two main components: artificial radiotherapy environment (ARTE) and optimal decision-maker (ODM). ARTE is composed of radio-biologically constrained transition functions for capturing patient's state dynamics, RT outcome estimator for predicting patient outcomes, and reward function for representing clinical goal. ODM is an AI agent trained using model-based deep reinforcement learning approach. ODM applies patient's pre- and mid-treatment information into ARTE, and based of the reward signals, can learn optimal decision-making process.

For improved explainability and transparency of AI's decision-making process, along with optimal recommendation, ARCliDS includes outcome space plot, reward signals, features distribution, and uncertainty estimates. ARCliDS presents its recommendations in the Outcome Space plot, spanned by TCP and NTCP, and contoured and colored with reward signal. Given a patient's information, the outcome space shows treatment outcomes and uncertainty estimate for a range of daily dose fractions [NSCLC: 1.5-4 Gy/fx; HCC: 1-15 Gy/fx]. The optimal dose recommendation and model uncertainty is marked with green diamonds. Additionally, since some of the multi-omics information is not used in the current clinical practice, ARCliDS provides patient-specific feature information in Population Distribution Plots. Knowing the patient's feature value and its relative position to the population helps end-users to visualize patient's "whereabouts".

## ARCliDS-NSCLC module

ARCliDS' NSCLC module was trained on a dataset obtained from UMCC 2007-123 phase II dose escalation clinical trial NCT01190527, where inoperable or unresectable non-small cell lung cancer patients were administered with 30 daily dose fractions[29]. The patients received roughly 50 Gy [Gray = J/Kg] equivalent dose in 2 Gy fractions (EQD2) in the evaluation phase and up to a total dose of 92 Gy EQD2 in the adaptation phase. The evaluation phase lasted for roughly two-thirds of the 6-week treatment period. For simplicity, the training dataset was divided into the evaluation phase of 20 fractions and the adaptation phase of 10 fractions. Two binary endpoints were considered for training: local control (LC) and radiation-induced pneumonitis of grade 2 or higher (RP2).

NSCLC patient's state was defined as the multi-omics features resulting from a multi-objective Markov Blanket feature selection process[56]. The study analyzed 297 multi-omics features and found 13 important features for predicting both LC and RP2. The selected features were cytokines: pretreatment interleukin 4 (pre-IL4), pre-IL15 and

slope of Interferon gamma-induced protein 10 (slope-IP10); Tumor PET imaging features/Radiomics: pretreatment Metabolic Tumor Volume (pre-MTV), relative difference (RD) of Gray-level size zone matrices (GLSZM)-large zone low gray-level (LZLGE) and RD-GLSZM-zone size variance (RD-GLSZM-ZSV); Dosimetry: Tumor gEUD and Lung gEUD; Genetics (single nucleotide polymorphism [SNP]): Cxcr1-Rs2234671, Ercc2-Rs238406, and Ercc5-Rs1047768; and MicroRNA: miR-191-5p and miR-20a-5p.

The reward function was adjusted to reflect the clinical trial goal of obtaining the population's local control rate of above 70% or higher and toxicity rate of 17.2% or lower, as follows:

$$r_{NSCLC} = \begin{cases} r+2, & if\ tcp > 0.70\ and\ ntcp < 0.172 \\ r+1, & if\ tcp > 0.50\ and\ ntcp < 0.50 \\ r, & otherwise \end{cases} \quad (1)$$

The AI gets a continuous baseline reward based on the monotonic reward function $r = tcp(1 - ntcp)$. One reward point is added if the AI decision leads to the computational goal of achieving $tcp > 50\%$ and $ntcp < 50\%$, and 2 points for clinical goal of $tcp > 70\%$ and $ntcp < 17.2\%$.

An 80-20 split 10-fold cross-validation using stratified shuffle was conducted to evaluate ARCliDS-NSCLC ARTE on the retrospective clinical decisions. The area under the receiver operating characteristics curve (AUROCC) for TCP was $0.73 \pm 0.15$ (mean ± SD) and NTCP was $0.79 \pm 0.17$. A self-evaluation schema based on radiobiological principle was designed to evaluate ODM. We found that ARCliDS-NSCLC ODM was able to reproduce 36% of the good clinical decisions, i.e., LC without RP2 and improve 74% of the bad clinical decisions.

## ARCliDS-HCC module

ARCliDS' HCC module was trained on a dataset obtained from the adaptive arm of the clinical trials NCT01519219, NCT01522937, and NCT0246083514, where hepatocellular carcinoma (HCC) patients received adaptive SBRT in a 3-2 split[30]. In the evaluation phase, patients received 3 high daily dose fractions followed by 11-month break, and in the adaptation phase, a suitable sub-population of the patients received 2 additional daily doses. Two binary endpoints were considered for training: local control (LC) and liver toxicity (LT) (≥ 2 points increase in Child-Pugh score during any point in the treatment.)

Similarly, HCC patient's state was defined from a multi-omics feature resulting from a human-in-the-loop based multi-objective Bayesian Network study[57]. The study analyzed 110 mutli-omics features and found 13 important features for predicting LC and liver toxicity. The selected features were clinical: sex, age, pretreatment cirrhosis status (pre-cirrhosis), pretreatment Eastern Cooperative Oncology Group Performance Status (pre-ECOG-PS), number of active liver lesions (active lesions), pretreatment albumin level (pre-albumin); Tumor PET Imaging: gross tumor volume (GTV) and liver volume minus GTV (Liver-GTV); Dosimetry: GTV gEUD and Liver-GTV volume; and cytokines/signaling molecule: relative difference of Transforming growth factor beta (RD-TGF-$\beta$), Cluster of Differentiation 40 receptor's Ligand (RD-CD40L), and Hepatocyte growth factor (RD-HGF).

The reward function was adjusted to reflect the high event rate of the HCC cohort, as follows:

$$r_{HCC} = \begin{cases} r+2, & if\ tcp > 0.90\ and\ ntcp < 0.25 \\ r+1, & if\ tcp > 0.50\ and\ ntcp < 0.50 \\ r, & otherwise \end{cases} \quad (2)$$

Again, the AI gets the baseline $r$ for all its decisions. One reward point is added if the AI decision leads to the computational goal and 2 points if AI can learn decisions that will lead to very high $tcp > 90\%$ and a low of $ntcp < 25\%$.

The AUROCC for ARCliDS'-HCC ARTE from an 80-20 split 10-fold cross-validation using stratified shuffle were $0.74 \pm 0.27$ for TCP and

$0.68 \pm 0.24$ for NTCP. Based on the radiological evaluation schema, ARCliDS'-HCC ODM was found to reproduce 50% of the good clinical decisions, i.e., LC without worsening of liver toxicity and improve 30% of the bad clinical decisions More details can be found in Niraula et al.'s work and its supplementary material[22].

We note that that although the multi-omics features used in our modeling came from Bayesian feature selection methods and were further vetted by domain experts, there is still room for inclusion of more features for modeling of diverse diseases and treatment response.

## Evaluation study design principle and implementation

This study is approved under IRB Moffitt Cancer Center- MCC# 20750. We designed two-phase evaluation modules, divided into Unassisted Phase and AI-assisted Phase, for ARCliDS-NSCLC and ARCliDS-HCC[22]. We chose a sequential design, where a unit of evaluation consisted of Unassisted Phase followed by AI-assisted phase for each patient. Doing so enabled us to perform a matched pair type of analysis in contrast to comparing whole distribution. In addition, we added text boxes for remarks and encouraged evaluators to provide comments which would provide valuable insight to their decision-making in the form of unstructured data.

To simplify the evaluation process, we designed the modules to be a stand-alone, interactive, and auto-saving web application; limited the patient count so that an evaluation could be completed in under an hour; developed tutorial videos; and conducted an initial pre-evaluation information session with the evaluators. To make it a stand-alone application that could operate without a standard treatment planner, we wrote Python scripts to preprocess treatment planning DICOM files into standard 3D NumPy arrays, then built a treatment plan viewer using Plotly library and incorporated it into the modules using the reticulate library. For interactive functions, we built the modules using R Shiny user-interface and Plotly graphing library. For auto-saving the evaluation, we linked the modules to google sheets using googlesheet4 library. For web accessibility, we hosted the modules in R shinyapps.io server. In addition to limiting the evaluation time, we included an online account system so that the evaluation could be completed in multiple sessions if needed. We developed tutorial videos explaining the motivation, functionality, and a brief overview of ARCliDS and evaluation modules, which was played to all evaluators during the initial pre-evaluation information session in addition to a demonstration of evaluation on a sample case.

To minimize biases, external influence, and systematic errors, we incorporated elements from randomization and isolation at different levels in this study. First, evaluation modules randomly initialized ordering of patients, so that each evaluator would interact with the same group of patients in a different ordering. Second, modules isolated the decision-making process by restricting evaluators to revisit the Unassisted Phase once they have seen the AI-recommendation and by deliberately excluding Unassisted Phase decisions from the AI-assisted Phase page. Third, we enrolled diverse evaluators from multiple institutions, different career levels and different specializations.

## Evaluation modules

**Workflow.** Fig. 1 summarizes the workflow between the main 3 components of the evaluation modules (EMs): User account and Data Frames, Unassisted page, and AI-Assisted page. The modules start at the Welcome Page which leads to either Login page for returning evaluators or Create Account page for new evaluators. All information is auto saved into Data Frames and depending on the status of the evaluator, they are prompted to either Unassisted page to start or the latest phase they left at.

In the Unassisted Phase, we present necessary clinical information, treatment plan, and images from the KBR-ART Evaluation Phase. Based on the provided information, the evaluators are asked to input

their decision for the following KBR-ART Adaptation Phase, their decision confidence level, and any remarks they may have. Once the evaluators are satisfied with their input, they can submit, which leads them to the AI-assisted Phase. In the AI-assisted Phase, we provide full access to ARCliDS. After seeing the ARCliDS' recommendation and outcome estimates, the evaluators are asked to re-enter their decision, decision confidence level, and their trust level on the AI Recommendation. Again, once the evaluators are satisfied with their input, they can submit, which completes the evaluation process for that patient. This process is repeated until the patient list is exhausted, which then leads them to the Exit Page which presents the input summary and a thank you message. In this study, we included 8 NSCLC patients and 9 HCC patients, which were randomly selected from the original training dataset based on availability of imaging information and also to limit the evaluation time to under one hour.

**User account, data frames, and tutorial video.** We built a rudimentary account system consisting of a Welcome page, Create Account page, Log In page, and Data Frame to allow users to finish the evaluation in multiple sessions if needed. The Welcome page contains web links11,12 to a tutorial video and ARCliDS manuscript13. The 10-minute-long tutorial video—created using PowerPoint and hosted in YouTube—contains description of KBR-ART; clinical trials and training dataset on which ARCliDS was trained7,9; ARCliDS architecture and its graphical user interface; and evaluation questionnaire. Two separate training videos were created for each of the two diseases: NSCLC and HCC.

The welcome page contains two push buttons for new and returning evaluators. The new evaluator button is linked to the Create Account page, where the new evaluators must input their name, level (physicians or resident), Affiliation, Specialization, Experience (number of years), unique user ID, and 8-digit PIN, to create a new account. The user ID is actively validated against the existing accounts and the 8-digit PIN is actively validated for the length. Once all the information is provided, clicking on the Start Evaluation button will save the information in the cloud Data Frame and take the evaluators to the Unassisted Phase page. The returning evaluator button is linked to the Log In page, where the returning evaluators must input their existing user ID and 8-digit PIN to continue their evaluation.

Data Frames are stored as google sheets. We use two data frames: the first to save all the evaluation material and the second to store the account information, validate the account information, randomly initialized patient ordering, and to keep track of the latest evaluated patient and the phase for each evaluator. The latter record is used as a guide for the returning users.

**Unassisted page.** In the Unassisted page, we recreated clinical workflow by presenting patient's information, Evaluation Phase treatment plan including PET/MRI images, and questionnaires as detailed in subsequent subsections.

**Patient Information.** A summary of the Patient information is listed in the supplementary information. In NSCLC module, we presented patient's sex, age, cancer stage, smoking history (binary), chronic obstructive pulmonary disease status (COPD, binary), cardiovascular disease status (CVD, binary), hypertension status (binary), histology (categorical), chemo status (binary), Karnofsky performance status (KPS), and gross tumor volume (GTV, in cc). In HCC module, we presented patient's sex, age, cirrhosis status (binary), number of active lesion, portal vein thrombosis (PVT) status (binary), number of pre-SBRT line of systemic therapies, number of pre-SBRT liver-directed therapy, presence of extrahepatic disease status (binary), number of prior liver occurrence, previous treatment status (binary), eastern cooperative oncology performance status (ECOG-PS), gross tumor volume (GTV, in cc), and Liver minus GTV (in cc). In addition, we presented pre-treatment and mid-treatment liver functions in terms of

Albumin level (g/DL), Bilirubin level (g/DL), ALBI score, and Child-Pugh (CP) score.

**Treatment Plan, PET, and MRI Viewers.** We designed evaluation modules as stand-alone applications for which we wrote Python scripts using pydicom and dicompylercore library to convert and co-register various DICOM files into NumPy arrays and then developed treatment plan viewers in R using Plotly library to independently incorporate them into the modules as shown in the supplementary information.

For each treatment plan we took patient's CT slices, RT structures, and 3D Dose Distribution DICOM files,

1. performed pixel to co-ordinate transformation,
2. selected overlapping region between CT and 3D Dose array,
3. enlarged and interpolated the 3D dose images to match CT resolution via scipy.ndimage.zoom function,
4. combined all RT structures into one 3D grid by,
   a. assigning unique integer value to the pixels of different structures and
   b. adding the structures together and saved all the transformed images into compressed 3D NumPy arrays.

In parallel, we extracted cumulative dose volume histogram (DVH) from 3D dose distribution DICOM files. In addition, to replicate the decision-making process in the adaptive RT clinical trials, we presented pre- and mid-treatment PET images in the NSCLC module and pre-treatment MRI in HCC module. We carried out the same procedure for registering PET and MRI images onto CT and RT Structure.

To reduce rendering time, we chose to show 2D slices of the compressed 3D NumPy arrays. To maintain the ease of viewing, we followed the standard procedure of showing images in Axial, Coronal, and Sagittal axis. In each viewer, in addition to native Plotly controls, we added three more controls:

1. slice number input box for selecting the slice number,
2. Dose/PET/MRI opacity slider for changing the intensity of the images,
3. Structure opacity slider for changing the intensity of the structure.

The intensity of the CT background is left fixed while discrete color bars were included for Structure names and continuous color bar for Dose value. We separately presented the treatment plan and PET/MRI images in two tabs. We added a separate viewer for DVH in the treatment plan, in which wanted/unwanted histogram can be selected/unselected using the native Plotly controls. In NSCLC module's PET tab, for an easy comparison, we added three axial viewers for pre-treatment PET and another three for mid-treatment PET in a side-by-side fashion. In HCC module's MRI tab, we included three pre-treatment MRI images similar to the dose plan. We modified the native Plotly zoom controls to maintain the original aspect ratio and also included an Enlarge button for every viewer for viewing the images in a large pop-up Modal Dialog Box.

**Questionnaire.** We asked the following two questions:

i. Unassisted Dose Decision: Recommend a dose adaptation value between 1.5 to 4.0 Gy/frac (NSCLC) | 1.0 to 15.0 Gy/Frac (HCC) that best fits the current patient.
ii. Decision Confidence Level: On a level of 0 (lowest) to 5 (highest), how confident are you in your decision?

In addition, we included a textbox for remarks. Once satisfied with the inputs, evaluator could then click on the next button to submit the evaluation and go to the AI-assisted page.

**AI-assisted page.** AI-assisted page consists of AI recommendation, outcome space, feature distribution, and questionnaires as shown in the supplementary information. We also included a few help-push buttons on the page.

**AI recommendation.** We presented recommendations from an ensemble of 5 AI models. The AI-recommendation is provided as mean ± sem of the daily dose fractionation for the KBR-ART adaptive phase. In addition, we provided corresponding total equivalent dose in 2 Gy fractions (EQD2)—a more clinically meaningful metric.

**Outcome Space.** We provided outcome estimation from an ensemble of 5 AI models in the outcome space spanned by TCP and NTCP corresponding to adaptive daily dose fractionation value ranging from 1.5 to 4.0 Gy/frac for NSCLC and from 1.0 to 15.0 Gy/frac for HCC. The outcome estimates are color-coded continuously from yellow (lowest dose) to red (highest dose) color. The outcome corresponding to the AI-recommendation are marked with the green diamond markers and the model uncertainty (sem) is given as error bars in both TCP and NTCP direction. The background of the outcome space is colored according to the AI reward function, $r = tcp(1 - ntcp)$, which is highest at clinically desired outcome, $(tcp, ntcp) = (1, 0)$.

**Feature Distribution.** For improving interpretability, we presented distribution plots for all patient's features, where feature values are presented with population density in the background to provide "whereabouts" about the patient. Each feature plot could be enlarged by clicking on the background. Feature descriptions table from the work of Niraula et al.[22] including weblink to relevant literature were provided in the help box.

**Questionnaire.** First, we re-asked the same two questions as the Unassisted phase as following:

i. AI-assisted Dose Decision: Having seen AI's prediction and recommendation, re-recommend a dose adaptation value between 1.5 and 4.0 Gy/frac (NSCLC) | 1.0 to 15.0 Gy/Frac (HCC) that best fits the current patient.
ii. Decision Confidence Level: On a level of 0 (lowest) to 5 (highest), how confident are you in your decision?

Second, to objectively quantify evaluators' trust level on ARCliDS recommendation, we asked following four multiple choice questions related to estimated outcome estimation and its associated uncertainty level and AI recommendation and its associated uncertainty level.

1. RT Outcome Estimation:

i. How does the model's estimation for outcomes for the dosage range seem to you? [1pt]

a. Reasonable     b. Unrealistic

ii. How does the range of uncertainty for outcomes seem to you? [1pt]

a. Reasonable     b. Unreasonable

2. AI Recommendation:

i. What is your view on the AI Recommendation Dose Range? [2 pts]

a. Agree     b. Disagree-Go Higher     c. Disagree-Go Lower

ii. How does the range of uncertainty for AI dosage recommendation seem to you? [1pt]

a. Reasonable     b. Unreasonable: Too Small     c. Unacceptable: Too Large

We chose the 4 multiple choice questions to total to 5 points, as same as the confidence level. We assigned 2 points to questions relating to evaluators view of the AI recommendation and 1 point to the rest. Only the first option, either Reasonable or Agree, carried the points and remaining options carried 0 points. Note that both the null options for AI-Recommendations, i.e. Disagree-Go Higher, Disagree-

Go Lower, Unreasonable: Too Small, and Unacceptable: Too Large received 0 points.

*Help:* We provided the following instruction/description in the help box.

AI Recommendation Trust Level (0-5): Calculated based on the following 4 questions.

1. RT Outcome Estimation
   i. Reasonableness of Estimation [1 pt]: Whether the outcome estimation is reasonable: if the shape of the path made by the tuple $(TCP, NTCP)$ makes sense, is physical, or is unrealistic or conflicting. For instance, both $TCP$ and $NTCP$ curves are assumed to monotonically increase with increasing dosage.
   ii. Uncertainty level [1pt]: Whether the Uncertainty envelope is reasonable or unacceptable: too small, too big, etc.
2. AI Recommendation
   i. Agreement with AI Recommended Dose Range [2 pts]: whether the evaluator agrees with the Recommended dose range i.e. $mean \pm sem$ or finds if the mean dose should be higher or lower.

      Help: Divide the TCP-NTCP outcome space into four quadrants which corresponds to four clinical outcome events. top-left: $(TC = 0, NTC = 1)$, top-right: $(TC = 1, NTC = 1)$, bottom-left: $(TC = 0, NTC = 0)$, and bottom-right: $(TC = 1, NTC = 0)$. Only the bottom-right: $(TC = 1, NTC = 0)$ is clinically desirable.
   ii. Uncertainty level [1pt]: See if the Uncertainty level (SEM) in AI Recommended dosage is acceptable or unacceptable. It could be clinically unacceptable if it's too large or unreasonable if it's too small.

## Module Deployment

We deployed both NSCLC and HCC evaluation modules on shinyapps.io servers. Prior to deployment, we obtained institutional cyber security clearance for hosting the modules externally, which took about 2 weeks from filling out an application to getting clearance but took was little over 4 months in total to navigate to the application in the first place.

We purchased the Standard shinyapps.io subscription capable of hosting a maximum of 5 instances (virtualized server, docker containers) at a time and at most 8 GB per instance, meaning that we had to take measures to limit module size to well below 8GB. We pre-processed DICOM images and saved 3D NumPy arrays in float16 data format, then we opted to show 2D image slices instead of 3D volumes, we precalculated and saved AI-recommendation and outcome estimation for all the patients, and we used Google Sheets to auto-save all the evaluation data instead of putting load to the server memory. Besides R, we had to set up Python interpreter during deployment.

## Enrollment, Training, and Evaluation

We advertised the study via a combination of group emails and in-person invitations in two institutions: Moffitt Cancer center and Michigan Medicine. Initially, a total of 20 volunteers showed interest and initial pre-evaluation information sessions were conducted, mostly in a one-on-one virtual meeting, two in-person meetings, and one group virtual meeting with three volunteers. Initially, we found a bug in the software, which was corrected, and the five evaluators who had taken the evaluation were requested to retake the evaluation to which only two out of five agreed, and thus the remaining 3 evaluations were discarded. Similarly, three evaluators didn't follow up and one evaluator only completed evaluation for one patient. In total, 13 evaluators completed the evaluation, and 4 evaluators volunteered to take both NSCLC and HCC evaluation, with a total of 17 completed evaluations (9 NSCLC, and 8 HCC).

A table of evaluator's information is presented in the supplementary information. The evaluators consisted of both physicians and residents from a variety of specializations and a range of experiences,

out of which two evaluators were residents during the initial training/meeting and physicians during the evaluation and were accordingly grouped in a special classification in the table. We have de-identified the name and institution and assigned numbers for those who completed the evaluation and alphabets for the rest.

For consistency, during the initial pre-evaluation information sessions, all evaluators were shown tutorial videos in addition to a demonstration[37,38]. The evaluators then completed the evaluation in their own time and technical support was provided whenever needed, either via zoom, email, or phone call. The evaluation study took a little over five months from the first advertisement to the completion of the last evaluation. Altogether, 144 set of decision samples were collected from 17 evaluations on 17 patients: for NSCLC, we collected 72 datapoints (8 patients × 9 evaluators) and for HCC, we collected 72 datapoints (9 patients × 8 evaluators). Data analysis was conducted after the end of the data collection period.

## Statistical analysis

**Matched Pair Randomization T-test.** Since both patients' and evaluators' sample size were small, we used randomization test to investigate the level of AI-influence in decision-making. Randomization test[40] is a non-parametric test that does not assume random sampling, normality of the population, or estimates population parameters such as mean and variance. In randomization, a test statistic is calculated from the observed data then compared with the distribution of the test statistics obtained from resampling the data as opposed to comparing with the standard distribution. Exchangeability under null is the central idea of randomization test, which roughly states that exchanging data under null should not alter the sample statistics significantly. We performed a two-tailed matched pair randomization t-test on: $H_0 : \Delta = 0, H_\alpha : \Delta \neq 0; \Delta = un - aia$, where the null hypothesis states that AI has no significant influence on the clinical decisions, or alternatively if AI fails to influence the decision, there should be no difference between unassisted decision and AI-assisted decision. In our case, the exchangeability under null translates to if AI had no influence, then an unassisted decision could have equally likely come from the group of ai-assisted decisions. The test was carried out following the codes from David C Howell[41].

**Correlation Analysis.** Besides hypothesis testing, we conducted correlation analysis between several observed and derived variables. We primarily used Spearman rank correlation ($\rho$) which quantifies the monotonic relationship between two variables. Unlike Pearson correlation ($r$), which quantifies the linear relationship, spearman correlation is less sensitive to extreme values (outliers). In addition, we report p-value for correlation coefficient, which corresponds to hypothesis test, $H_0 : \rho = 0, H_\alpha : \rho \neq 0;$ where the null hypothesis states that the correlation does not significantly differs from zero.

In contrast, we use Pearson correlation and scatter plot to investigate the hypothesis $H_0 : un = aia, H_\alpha : un \neq aia$ and $H_0 : un\ conf = aia\ conf, H_\alpha : un\ conf \neq aia\ conf$. We know that, under null, $un$ and $aia$ should show perfect (or near perfect) correlation and the tuple $(un, aia)$ should distribute close to the $un = aia$ line. Additionally, under null, the best linear fit line to the data should coincide with the null hypothesis line, $un = aia$. Same applies to the tuple $(un\ conf, aia\ conf)$.

**Derived Quantities.** In addition to the observed variables such as $un, aia, un\ conf, aia\ conf$, and *AI trust*, we derived number of quantities to investigate collaborative decision-making process.

1. *Decision adjustment frequency* is the number of cases where the decisions were adjusted after AI access, i.e. $\sum_{i=1}^{n} \delta_{un_i aia_i}$, where $\delta_{un_i aia_i} = 0$ if $un_i = aia_i$ else 1. Decision adjustment frequency is zero if all unassisted decisions are pairwise equal to AI-assisted decision.

2. *Decision adjustment level* is the difference between AI-assisted and Unassisted decision, measured in Gy/fx, i.e. $(aia - un)$.

3. *Dissimilarity in decision-making with AI* is the difference between AI recommendation and Unassisted decision, measured in Gy/fx, i.e. $(ai - un)$.

4. *Agreement with AI* is the additive inverse of the absolute difference between AI-assisted Decision and AI recommendation, i.e. $-|aia - ai| \in$ . The level of agreement peaks at 0 which corresponds to the absence of difference between *aia* and *ai*, and decreases when the difference between *aia* and *ai* increases in either direction.

5. *Closeness to Standard of Care* is the additive inverse of the absolute difference between decision and SOC, $-|d - SOC| \in (-\infty, 0]$, where $d \in \{un, aia\}$. Closeness peaks at 0 when $d = SOC$ and decreases when the difference between decision and SOC increases in either direction.

**Intraclass correlation coefficient.** We performed a concordance analysis on the decisions by comparing the Intraclass correlation coefficient (ICC) of *un* and *aia*. We chose McGraw and Wong's formulation of ICC for this study and assumed a two-way random effect linear model following the fact that both patients and evaluators were chosen at random from a larger pool, i.e. $d_{ij} = \mu + a_i + b_j + ab_{ij} + e_{ij}, d \in \{un, aia\}, i \in \{1, \dots, n\}, j \in \{1, \dots, k\}$, where *n* in the number of patients, *k* is the number of evaluators, $\mu$ is the population mean, $a_i$ is the inter-patient heterogeneity, *k* is inter-evaluator (physician) variability, $ab_{ij}$ is the patient–evaluator interaction variability and $e_{ij}$ is the random error. For completeness we calculated both ICC types: Consistency (C) and Absolute Agreement (A); for both units: Single rater (1) and Average rater (k), resulting in four combinations: $ICC(C, 1)$, $ICC(A, 1)$, $ICC(C, k)$, and $ICC(A, k)$. ICC type Consistency measures the symmetric differences between the decisions of the evaluators, and Absolute Agreement measures the absolute differences, making the latter a stricter form of coefficient. Similarly, ICC unit Single rater corresponds to using the decision from a single evaluator as the basis for measurement and ICC unit Average corresponds to using the average decision from *k* evaluators. The absolute value of ICC is study-dependent and hard to interpret on its own. However, comparing ICC of pairwise variables from the same study under identical conditions is meaningful. Thus, regardless of the strength of ICC, we can draw a conclusion on the inter-evaluator agreement by comparing ICC between decisions made by the same group of evaluators without and with AI-assistance. We used irr R-package to compute the ICCs.

**Toxicity Free Local Control Scoring Schema.** In the absence of ground truth, we analyzed the adjusted decision $(un \neq aia)$ based on a scoring schema $TCP(1 - NTCP) \in [0, 1]$, which reflects the clinical goal of achieving toxicity free local control. Mathematically, the scoring schema is the likelihood of achieving tumor control without a normal tissue complication $(1 - NTCP)$. It has a maximum value of 1 for ideal outcome of $(tcp, ntcp) = (1, 0)$ and minimum value of 0 for dose limiting factor $ntcp = 1$. In addition, the scoring function is a quadratic function having a higher sensitivity than a probability metric, for instance, the scoring function for $TCP$ and $NTCP$ in percentages would be $TCP(100 - NTCP)$ and would range from 0 to 10,000, thus we use four decimal places for analysis.

### Reporting summary
Further information on research design is available in the Nature Portfolio Reporting Summary linked to this article.

## Data availability
The source data (evaluator's decision data) are provided with this paper. IRB restrictions apply to the patient's clinical data and are not publicly available. For this study, the data were obtained from the University of Michigan under data sharing protocol. To obtain the data, a formal data sharing application must be submitted to the Department of Radiation Oncology, University of Michigan. Source data are provided with this paper.

## Code availability
ARIiCDS is an intellectual property of Moffitt Cancer Center and University of Michigan and thus restrictions apply to the source codes. A detailed description of the underlying algorithms has been published in https://doi.org/10.1038/s41598-023-32032-6. An online version of the ARCliDS is available at https://arclids.shinyapps.io/ARCliDS/; NSCLC evaluation module is available at https://arclids.shinyapps.io/Eval_NSCLC_v2/; and HCC evaluation module is available at https://arclids.shinyapps.io/Eval_HCC_v2/.

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

## Acknowledgements

This work was partly supported by the National Institute of Health (NIH) grant R01-CA233487 and its supplement.

## Author contributions

IEN conceived the study. KCC, IDD, JJ, MMM, RKTH, and IEN supervised the project. DN, IDD, BDG, JJ, and IEN designed the study. DN designed, developed, and deployed the ARCliDS software and the evaluation modules. DN and JBJ collected and curated patient data. KCC, JBJ, IDD, JJ, MMM, RKTH, and IEN evaluated the module design. DN developed the tutorial videos. DN, KCC, TJD, and IEN conducted evaluator searches. DN, KCC, and IEN conducted initial pre-evaluation information sessions with potential evaluators. AKB, TJD, KCC, MBD, JMF, CLL, SRM, MNM, RFP, SNR, AR, JFTR, and HHMY participated in the study and critically

evaluated the module. DN administered the evaluation study and provided tech support as needed. DN analyzed the results. IDD, BDG, JJ, and YL provided feedback on the quantitative analysis. All authors carefully analyzed the methods and results. DN drafted the manuscript, and all authors critically reviewed and contributed to the final version.

## Competing interests

DN, KCC, IDD, JBJ, JJ, YL, RKTH, AKB, MPD, JMF, CLL, SRM, MNM, RFP, SNR, and AR have no conflicting interests. BDG reports fees unrelated to this work from Sure Med Compliance and Elly Health. MMM reports research funding from Varian, a licensing agreement with Fuse Oncology, and serves in the AAPM Board of Directors and is the Co-Director of MROQC, funded by BCBSM. TJD is a member of the National Comprehensive Cancer Network (NCCN) NSCLC panel. JFTR reports stock ownership and leadership in Cvergenx, Inc. He reports IP and royalty rights in RSI, GARD, RxRSI. HHMY reports funding or fees unrelated to this work from the National Institute of Health, UpToDate, Novocure and Bristol-Myers Squib. IEN is on the scientific advisory of Endectra, LLC., co-founder of iRAI LLC, deputy editor for the journal of Medical Physics, co-chief editor of British Journal of Radiology (BJR)-AI and receives funding from the National Institute of Health (NIH), foundations, and Department of Defense (DoD). A PCT patent application for ARCliDS has been filed. Patent Title: Adaptive radiotherapy clinical decision support tool and related methods, Patent Applicant: H Lee Moffitt Cancer Center IP office in conjunction with University of Michigan IP office. Inventors: DN, IEN, RKTH, Wenbo Sun, JJ, IDD, KCC, MMM, and JBJ. Application Number: US2023/075004. Status of Application: Pending. Specific aspect of manuscript covered in patent application: The patent covers the underlying model-based decision-making framework of ARCliDS.
