## [Transparent Peer Review file · Nature Communications]

Intricacies of Human-AI Interaction in Dynamic Decision-Making for Precision Oncology

Corresponding Author: Dr Dipesh Niraula

Version 0:

Reviewer comments:

Reviewer #1

(Remarks to the Author)

After thoroughly reviewing the point-by-point responses from the authors, I still have concerns that the limited number of datasets is insufficient for such a challenging field as personalized radiotherapy. Importantly, the evaluation lacks fairness and generalization, and the AI model's focus is not comprehensive.

My specific comments are as follows:

[i] Limited dataset for such a novel and challenging field: Although this study focuses on a novel field, personalized radiotherapy for Human-AI interaction, the required continuous decisions and the absence of ground truth make it more difficult than traditional diagnostic AI fields. Thus, the current dataset size is likely far from sufficient to provide convincing conclusions in this new and challenging area.

[ii] Lack of generalization in evaluation: As detailed in section S3.1, the evaluated patients were all randomly selected from the original training dataset, which lacks generalization evaluation. This evaluation process may lead to overclaiming some conclusions. Even if using AI in treatment processes presents ethical risks, conducting experiments under a validation set of the same dataset can significantly improve this issue.

[iii] Nature of the regression problem: I still believe the problem being studied is fundamentally a regression problem. Reinforcement learning or RLHF is used here merely as a strategy to improve regression performance, even without ground truth. The two "doesn't" situations mentioned in the response to Comment C.1 refer to the training strategy of reinforcement learning but do not change the core regression nature of the AI model.

[iv] Non-comprehensive consideration for the AI model: Despite the evaluation procedure using patients' multi-modal information, the AI model's input features are limited (only 13) and not comprehensive, even when considering a Bayesian feature selection strategy. Furthermore, the utility of an AI model plays a critical role in human-AI collaboration. Different outcomes may arise when collaborating with a simple model versus a more advanced model, such as ChatGPT.

(Remarks on code availability)

Reviewer #2

(Remarks to the Author)

I appreciate the authors' responses. As noted in my original review, I believe the study has merit despite some obvious limitations, which the authors have acknowledged. While no additional experiments or data were added, which is understandable given the complex data collection procedure, the responses to the reviewers' queries are mostly satisfactory, and the changes made are appreciated. However, some minor points still need to be addressed.

Reviewer2, comment #3a:

The authors clarified in their response that their use of "adaptive" refers specifically to the adaptation of treatment based on clinical data, particularly longitudinal multi-omics data. Other interpretations of adaptive, such as considering patients' or physicians' preferences, were not included in this study. However, fully embracing precision oncology may necessitate a broader concept of "adaptive." In the revised version of the manuscript, the authors highlight several topics for future research on Human-AI collaboration in medical decision-making. Nevertheless, they do not acknowledge other attempts at a more general approach to adaptive treatment decisions in oncology with AI support beyond supervised learning. Notably, Barata et al. recently addressed some of these topics by studying the selection of therapeutic decisions for skin cancer treatment, comparing AI support via supervised learning and reinforcement learning while considering physicians' preferences. This approach should be mentioned to provide a more comprehensive context for ongoing research in this field.

Reviewer 2, comment #3b: The sample size-related issues have been acknowledged by the authors. My only further concern is that this should be made as transparent as possible in the figures. In Figure 2C, D, G, and H, for example, the sample size is somewhat concealed by reporting only percentages instead of actual numbers. This should be corrected by providing the numbers for each bar alongside the percentages, with the total number of decisions, patients and raters included in the legend.

Reviewer 2, comment #4: Although I understand that comparing dose administration with and without AI support in two groups presents significant challenges and is beyond the current scope of the study, this clinical test will ultimately be necessary to approve AI support for this purpose. The authors argue that inter-patient heterogeneity makes such studies challenging. However, this is a common issue in clinical medicine and randomization is designed to address inter-patient heterogeneity by ensuring that both, known and unknown patient characteristics, are evenly distributed among study groups.

Reviewer 2, comment #7: The argument that reduced ICC is more related to the uniformity of decision-making at the physician level and less at the patient level slightly misses the point. Uniformity in decision-making among physicians can be undesirable if the information informing their decisions is misleading. In such cases it would be better to have less uniformity to prevent collective errors. If everyone is influenced by the same type of misinformation, it can be detrimental. This is a potential risk of AI support that needs to be addressed. The risk might be outweighed by the benefits of AI-support, but this is not guaranteed.

Reviewer 3, comment and suggestion 1:

The results show that most decisions were adjusted to achieve higher TCP in NSCLC patients and lower NTCP in HCC patients. I still do not understand why this occurs, and there is no clear explanation for this behavior in the discussion (perhaps I missed it). Could it be that the evaluators use different trade-offs between TCP and NTCP for NSCLC and HCC patients? If so, what are the reasons for this? If there are valid reasons for adjusting the trade-off according to the disease, why not incorporate this into the AI training?

Harald Kittler

(Remarks on code availability)

I attempted to review the code, but the provided instructions to access the repository did not work. Specifically, I was asked to verify my identity to access the given Gmail address.

Reviewer #5

(Remarks to the Author)

Study Summary

This manuscript reports statistics on the use of an artificial intelligence clinical decision support system (AI-CDSS) by 13 radiation oncology resident and attending physicians from two medical institutions.

The AI-CDSS, called ARClIDS, provides adaptive RT dose and fraction suggestions given inputted features of non-small cell lung cancer (NSCLC) and hepatocellular carcinoma (HCC) cases. ARClIDS suggestions are based on training of its AI model on adaptive RT clinical trial data. Training of ARClIDS was completed and reported in previously published work.

Key Conclusions

By analyzing 17 completed evaluations (9 NSCLC, and 8 HCC) for each physician, the authors identified several key trends, including but not limited to:

- (1) Evaluators tended to adjust their decision closer towards AI recommendation.
- (2) Evaluators followed AI's recommendation if they agreed with that recommendation.
- (3) Evaluators were generally more confident in prescribing dose fractionations more similar to those used in standard clinical practice.
- (4) Majority of decisions were adjusted to achieve higher tumor control in NSCLC and lower normal tissue complications in HCC.

The manuscript finishes with a discussion of study limitations and future directions. Most importantly, the authors discussed the small sample size analyzed in the study and the lack of stratified sub-group analyses.

Quality of Evidence

1. The authors used appropriate statistical tests and evaluation methods.
2. Unfortunately, the small sample size and lack of sub-group analyses leaves a reader uncertain if the results have both internal and external validity.
3. The authors provided access to code on GitHub and the ARCLiDS user interface, which works well enough to demonstrate the AI system functions.

Significance to the Field

1. The authors emphasize that the results presented here are novel because it is a first-time evaluation of a real-world AI-CDSS that provides a continuous output instead of categorical one. While this statement is true, it may be viewed by some readers as a subtle distinction of novelty.
2. The authors appropriately benchmark their results to previous analyses of categorical AI-CDSS from different specialties.
3. I am a radiation oncologist, and I greatly appreciate this study and its contribution to the field. However, readers from other disciplines may view the work as too narrow-focused and not broadly applicable to other medical specialties.
4. While this work does an excellent job at untangling the nature of the AI-user interaction, it does not examine its consequence on patient outcomes. Although this potential analysis is mentioned in future directions, it is a necessary piece to demonstrate substantial enough significance for publication in this journal.
5. One key element missing from experimental design is whether or not the treating radiation oncologist is likely to change their case approach if and when AICDSS provides suggestions that are far afield from reasonable. By 'spiking' in bad suggestions purposely, are the radiation oncologists editing the case still likely to change their approach in the direction of the AICDSS recommendation? If the model performs poorly, how likely is the treating physician to in turn take an approach that could lead to patient harm? This is a crucial element and a great fear of over-reliance on AI that this study could have investigated but we are left guessing - possibly?

(Remarks on code availability)

Reviewer #6

(Remarks to the Author)

(Remarks on code availability)

The authors provided access to code on GitHub and the ARCLiDS user interface, which works well enough to demonstrate the AI system functions. I did not personally attempt to install and run the code on my local device.

Version 1:

Reviewer comments:

Reviewer #1

(Remarks to the Author)

This work discussed an important question of using AI for decision-making despite the small data size. The authors have addressed all my concerns. Therefore, I recommend accepting this paper for possible publication in Nature Communications.

(Remarks on code availability)

Reviewer #2

(Remarks to the Author)

The authors have adequately addressed all points raised by the reviewers

(Remarks on code availability)

RESPONSE TO REVIEWERS' COMMENTS

Reviewer #1 (Remarks to the Author):

After thoroughly reviewing the point-by-point responses from the authors, I still have concerns that the limited number of datasets is insufficient for such a challenging field as personalized radiotherapy. Importantly, the evaluation lacks fairness and generalization, and the AI model's focus is not comprehensive. My specific comments are as follows:

[i] Limited dataset for such a novel and challenging field: Although this study focuses on a novel field, personalized radiotherapy for Human-AI interaction, the required continuous decisions and the absence of ground truth make it more difficult than traditional diagnostic AI fields. Thus, the current dataset size is likely far from sufficient to provide convincing conclusions in this new and challenging area.

We agree that limited data presents a challenge for modeling in personalized radiotherapy and precision oncology. However, it is representative of the current status in the field. This is an emerging field, and standard clinical practice hasn't yet fully incorporated multi-omics-based adaptive radiotherapy, these are steps towards this goal. Additional clinical trials are ongoing and larger data collection is taking place. A single clinical trial takes at least 5-7 years to complete because treatment outcomes (local tumor control, side-effects) are measured up to 5 years; and one clinical trial generates about ~100 data samples after data preprocessing. So, the datasets are not large enough compared to other domains.

Nevertheless, we have put effort into overcoming limitations that we recognize as well. We present AI model uncertainties (outcomes and recommendations) by repeating experiments: 5 models for RTOE and ODM each. The uncertainty is then derived from the model output distribution. For this study, we obtained datasets from two groups of clinical trials (lung, liver). The qDRL model prior to ARCLiDS (Ref 21) was externally validated on the NRG RTOG0617 dataset, whose data agreement has now expired. Additionally, from 2020 to 2022, we submitted 3 NRG Oncology ancillary project applications with follow-up review, requesting the RTOG 1106 dataset (mid-treatment PET-based adaptive RT), which were declined pending the publication of their primary study. We will submit a follow-up application request soon after its publication for further validation of the presented models taking also into account the multi-institutional nature of such dataset.

[ii] Lack of generalization in evaluation: As detailed in section S3.1, the evaluated patients were all randomly selected from the original training dataset, which lacks generalization evaluation. This evaluation process may lead to overclaiming some conclusions. Even if using AI in treatment processes presents ethical risks, conducting experiments under a validation set of the same dataset can significantly improve this issue.

This is related to the same dataset limitation issue. Due to the unavailability of an external dataset with similar characteristics, we randomly selected patients from the study cohort. However, since the primary purpose of this study is to evaluate AI's influence on physicians' clinical decisions rather than to assess AI's performance, which has been previously presented [Ref 22], there is no data/information leakage by including patients from the original dataset because the evaluators had never seen the patients before. The retrospective outcome and clinical decision were withheld from the evaluators. In other words, the evaluators don't know if the AI recommendation and projected outcome predictions are accurate, i.e., there is no information leakage into the evaluation. Furthermore, in this study, we didn't find any evidence of overreliance

on AI. However, we acknowledge that the inclusion of external validation should help to reduce potential bias. In the revised article, we have now acknowledged this potential drawback.

Other factors are, 1) the Human-AI behavior observed in this study has been compared with other published studies, and we have found a lot of similarities [Ref. 15, 45, 46,47]. (2) Ultimately, estimating actual behavior will require conducting Human-AI interaction study in a prospective setting. As McIntosh et al. [53] observed, we believe that physicians' behavior will likely be more conservative in real settings, as mentioned in the discussion section.

[iii] Nature of the regression problem: I still believe the problem being studied is fundamentally a regression problem. Reinforcement learning or RLHF is used here merely as a strategy to improve regression performance, even without ground truth. The two "doesn't" situations mentioned in the response to Comment C.1 refer to the training strategy of reinforcement learning but do not change the core regression nature of the AI model.

We agree that our decision-making problem can be viewed as a regression problem under the definition: the relationship between dependent variable (decision) and independent variables (features). However, the role of outcome slightly complicates the consideration, thus interpreting the root mean square difference between retrospective decisions and AI recommendations needs special consideration, which is why we came up with the Self-Evaluation Scheme. Concretely, given a tuple (X, d) where X is the patient feature and d is the retrospective decision, in regression, the goal would be to learn/train a model F that would output $F(X) = d'$ that is close to d , which is done by minimizing loss function such as mean squared error in between $F(X)$ and d . However, in our case, we want to learn only the good decision (d_g) and want to avoid learning bad decisions (d_b). The good and bad decisions are defined by the retrospective competing risk outcome characterized by two events: local tumor control (TC) and radiation-induced normal tissue complication (NTC). Good decisions correspond to $TC = 1$ and $NTC = 0$, and the remaining three clinically undesirable scenarios, $(TC, NTC) = \{(1,1), (0,0), (0,1)\}$, originates from bad decisions. So essentially, we first develop a radiotherapy outcome estimator (ROTE), which can estimate the outcome probability of dose decisions: TCP and $NTCP$. Using the estimate, we define a reward signal [$R = TCP (1 - NTCP)$], which encompasses clinical objective, i.e., R is maximum when $TCP = 1$ and $NTCP = 0$. The reward function is then used to differentiate between decisions (d) and only then the problem translates to a regression where we maximize cumulative R , which in our case was done by Q-learning.

[iv] Non-comprehensive consideration for the AI model: Despite the evaluation procedure using patients' multi-modal information, the AI model's input features are limited (only 13) and not comprehensive, even when considering a Bayesian feature selection strategy. Furthermore, the utility of an AI model plays a critical role in human-AI collaboration. Different outcomes may arise when collaborating with a simple model versus a more advanced model, such as ChatGPT.

We agree that although the multi-omics features used in our modeling came from Bayesian feature selection methods and were further vetted by domain experts, there is still room for inclusion of more features for modeling of diverse diseases and treatment response. We have now explicated this point in the Methods section.

Reviewer #2 (Remarks to the Author):

I appreciate the authors' responses. As noted in my original review, I believe the study has merit despite some obvious limitations, which the authors have acknowledged. While no additional experiments or data were added, which is understandable given the complex data collection procedure, the responses to the reviewers' queries are mostly satisfactory, and the changes made are appreciated. However, some minor points still need to be addressed.

Reviewer2, comment #3a:

The authors clarified in their response that their use of "adaptive" refers specifically to the adaptation of treatment based on clinical data, particularly longitudinal multi-omics data. Other interpretations of adaptive, such as considering patients' or physicians' preferences, were not included in this study. However, fully embracing precision oncology may necessitate a broader concept of "adaptive." In the revised version of the manuscript, the authors highlight several topics for future research on Human-AI collaboration in medical decision-making. Nevertheless, they do not acknowledge other attempts at a more general approach to adaptive treatment decisions in oncology with AI support beyond supervised learning. Notably, Barata et al. recently addressed some of these topics by studying the selection of therapeutic decisions for skin cancer treatment, comparing AI support via supervised learning and reinforcement learning while considering physicians' preferences. This approach should be mentioned to provide a more comprehensive context for ongoing research in this field.

-We thank the reviewer for the reference. We have now acknowledged and referenced the similarity and stated the differences with Barata et al.'s approach in the discussion section.

Reviewer 2, comment #3b: The sample size-related issues have been acknowledged by the authors. My only further concern is that this should be made as transparent as possible in the figures. In Figure 2C, D, G, and H, for example, the sample size is somewhat concealed by reporting only percentages instead of actual numbers. This should be corrected by providing the numbers for each bar alongside the percentages, with the total number of decisions, patients and raters included in the legend.

- We have now included the absolute values next to the corresponding percentages in Figure 2c, 2d, 2g, and 2f. Additionally, the number of subgraphs presented in Figure 2a, 2b, 2e, and 2f represents the number of evaluators and patients.

Reviewer 2, comment #4: Although I understand that comparing dose administration with and without AI support in two groups presents significant challenges and is beyond the current scope of the study, this clinical test will ultimately be necessary to approve AI support for this purpose. The authors argue that inter-patient heterogeneity makes such studies challenging. However, this is a common issue in clinical medicine and randomization is designed to address inter-patient heterogeneity by ensuring that both, known and unknown patient characteristics, are evenly distributed among study groups.

- Yes, we agree that comparing outcomes of two groups: without (control group) and with (test group) AI support will be necessary to validate AI's utility before clinical application. We have acknowledged that in the Discussion section.

Reviewer 2, comment #7: The argument that reduced ICC is more related to the uniformity of decision-making at the physician level and less at the patient level slightly misses the point. Uniformity in decision-making among physicians can be undesirable if the information informing their decisions is misleading. In such cases, it would be better to have less uniformity to prevent collective errors. If everyone is influenced by the same type of misinformation, it can be detrimental. This is a potential risk of AI support that needs to be addressed. The risk might be outweighed by the benefits of AI-support, but this is not guaranteed.

Thank you for the clarification. We agree that a higher ICC value denotes uniformity of decision-making due to higher AI-influence on decisions, which can be harmful if the influence came from misinformation. We have added a clarification in the discussion addressing the raised concern. To add to the point, such risk warrants an increase in model transparency, interpretability, and explainability, such that human experts can clearly weigh in the AI-recommendation before making a final decision (selection of a decision option) without over-relying on the AI.

Reviewer #2 (Remarks on code availability):

I attempted to review the code, but the provided instructions to access the repository did not work. Specifically, I was asked to verify my identity to access the given Gmail address.

Our apologies: the instructions could have been more straightforward. We have now submitted a zip folder with the source codes. The folder contains five subfolders, as listed below:

1. DipeshNiraula/arclids-training:
 - a. Training files for ARCLiDS previous study
2. DipeshNiraula/arclids:
 - a. Codes for Online App previous study
 - b. ReadMe.md has files description.
 - c. repo also contains Arclids shiny app link
 - d. WebLink for user interface: <https://arclids.shinyapps.io/ARCLiDS/>
3. DipeshNiraula/arclidsEvaluationModule-NSCLC
 - a. Code for Online NSCLC Evaluation Module used for this study.
 - b. ReadMe.md has file description.
 - c. repo also contains shiny app link and youtube tutorial video.
 - d. WebLink: https://arclids.shinyapps.io/Eval_NSCLC_v2/
 - e. Tutorial Video: https://youtu.be/R_1D3ZvUt1E?si=gZCxNsd9lwigHNf9
4. DipeshNiraula/arclidsEvaluationModule-HCC
 - a. Code for Online NSCLC Evaluation Module used for this study.
 - b. ReadMe.md has files description.
 - c. Repo also contains shiny app link and youtube tutorial video.
 - d. WebLink: https://arclids.shinyapps.io/Eval_HCC_v2/
 - e. Tutorial Video: <https://youtu.be/WYS1WoAwhr0?si=-fYne9ExNJ3qcT4T>
5. DipeshNiraula/DICOM_to_numpy
 - a. Code for converting and co-registering various Treatment Planning DICOM files into NumPy arrays.

Note: For privacy protection concerns, the repo does not contain patient data so running the Evaluation Module locally might raise errors. Please refer to the online app.

Reviewer 3, comment and suggestion 1:

The results show that most decisions were adjusted to achieve higher TCP in NSCLC patients and lower NTCP in HCC patients. I still do not understand why this occurs, and there is no clear explanation for this behavior in the discussion (perhaps I missed it). Could it be that the evaluators use different trade-offs between TCP and NTCP for NSCLC and HCC patients? If so, what are the reasons for this? If there are valid reasons for adjusting the trade-off according to the disease, why not incorporate this into the AI training?

This is an excellent comment. For NSCLC, we observed that 76% of decision adjustments resulted in an increase in TCP and NTCP and for HCC, we observed that 74% of decision adjustments resulted in decrease in TCP and NTCP. Based on the rationale that the clinical goal of RT is to increase TCP while decreasing NTCP, we deduced that majority (76%) of NSCLC decisions were adjusted to increase TCP while majority (74%) of for HCC decisions were adjusted to decrease the NTCP. We have now added further clarification in the Results section.

We think that model-specific behavior change can be attributed to the difference in treatment options, organ and cancer type, and treatment protocols. NSCLC patients were administered RT in 30 fractions at the rate of 2 Gy/frac, while HCC patients were administered SBRT in 5 fractions at the rate of 10Gy/frac. The higher rate of dose causes more concern for the toxicities. Another difference is the event rate of the cohort: 95 out of 99 HCC patients showed local control (Table S10 in Ref 22). which is much higher than in NSCLC. Such underlying distribution was learned by the AI outcome prediction model for HCC, which made a high TCP projection for all patients and only changed slightly for the full range of dose decision values. Note that such behavior is similar to a type of AI bias shown by language models where AI learns slightly differently for underrepresented samples.

Additionally, for NSCLC, pneumonitis was the primary concern of toxicity, and for HCC, liver function, which was measured in the Child-Pugh score. These differences may have led to different decision adjustment behaviors, in particular, increasing TCP for NSCLC and decreasing NTCP for HCC.

We have incorporated the trade-offs for NSCLC and HCC through a reward function. For NSCLC, to reflect the clinical trial goal of obtaining the population local control rate of above 70% or higher and toxicity rate of 17.2% or lower, the reward function was defined as follows:

$$r_{NSCLC} = \begin{cases} r + 2, & \text{if } tcp > 0.70 \text{ and } ntcp < 0.172 \\ r + 1, & \text{if } tcp > 0.50 \text{ and } ntcp < 0.50 \\ r, & \text{otherwise} \end{cases}$$

The AI gets a continuous reward based on the monotonic reward function $r = tcp(1 - ntcp)$. One reward point is added if the AI decision leads to the computational goal of achieving $tcp > 50\%$ and $ntcp < 50\%$, and 2 points for the clinical goal of $tcp > 70\%$ and $ntcp < 17.2\%$. For HCC, the high event rate was incorporated as follows:

$$r_{HCC} = \begin{cases} r + 2, & \text{if } tcp > 0.90 \text{ and } ntcp < 0.25 \\ r + 1, & \text{if } tcp > 0.50 \text{ and } ntcp < 0.50 \\ r, & \text{otherwise} \end{cases}$$

Again, the AI gets the baseline r for all its decisions. One reward point is added if the AI decision leads to the computational goal and 2 points for the decision lead very high $tcp > 90\%$ and a low of $ntcp < 25\%$. To clarify this point, we have added the above description in the Methods section.

Reviewer #5 (Remarks to the Author):

Study Summary

This manuscript reports statistics on the use of an artificial intelligence clinical decision support system (AI-CDSS) by 13 radiation oncology residents and attending physicians from two medical institutions.

The AI-CDSS, called ARcliDS, provides adaptive RT dose and fraction suggestions given inputted features of non-small cell lung cancer (NSCLC) and hepatocellular carcinoma (HCC) cases. ARcliDS suggestions are based on training of its AI model on adaptive RT clinical trial data. Training of ARcliDS was completed and reported in previously published work.

Key Conclusions

By analyzing 17 completed evaluations (9 NSCLC, and 8 HCC) for each physician, the authors identified several key trends, including but not limited to:

- (1) Evaluators tended to adjust their decision closer towards AI recommendation.
- (2) Evaluators followed AI's recommendation if they agreed with that recommendation.
- (3) Evaluators were generally more confident in prescribing dose fractionations more similar to those used in standard clinical practice.
- (4) Majority of decisions were adjusted to achieve higher tumor control in NSCLC and lower normal tissue complications in HCC.

The manuscript finishes with a discussion of study limitations and future directions. Most importantly, the authors discussed the small sample size analyzed in the study and the lack of stratified sub-group analyses.

Quality of Evidence

1. The authors used appropriate statistical tests and evaluation methods.

We thank the reviewer for the positive comment.

2. Unfortunately, the small sample size and lack of sub-group analyses leaves a reader uncertain if the results have both internal and external validity.

ARcliDS has only been validated on internal datasets [Ref 22]; however, the underlying AI decision model was also externally validated in the RTOG0617 dataset in Ref 21.

This study focuses on the clinicians and their behavior change, who had never seen the patients before. Additionally, the retrospective decision and outcome were withheld. Thus, this study itself is an external validation, albeit on a small sample size.

3. The authors provided access to code on GitHub and the ARcliDS user interface, which works well enough to demonstrate the AI system functions.

We thank the reviewer for the positive comment.

Significance to the Field

1. The authors emphasize that the results presented here are novel because it is a first-time evaluation of a real-world AI-CDSS that provides a continuous output instead of categorical one. While this statement is true, it may be viewed by some readers as a subtle distinction of novelty.

In our literature search, we found Human-AI interaction studies for diagnostic AI's, where the endpoints (e.g., disease classification) were readily available. Our case is unique because we

don't have endpoint (treatment outcome for all possible dosages), thus, we had to define several derived quantities to study behavioral change with AI-assistance. Our methodologies of analyzing Human-AI collaborative decision-making will provide a framework for future Human-AI interaction studies in therapeutics where the endpoints are generally unavailable. We have emphasized this in the Discussion.

2. The authors appropriately benchmark their results to previous analyses of categorical AI-CDSS from different specialties.

We thank the reviewer for the positive comment.

3. I am a radiation oncologist, and I greatly appreciate this study and its contribution to the field. However, readers from other disciplines may view the work as too narrow-focused and not broadly applicable to other medical specialties.

We thank the reviewer for the positive comment. We investigated and found two levels of behavioral change: model-agnostic and model-specific. We have now clarified that the model-agnostic behavioral change should translate to other medical domains.

4. While this work does an excellent job at untangling the nature of the AI-user interaction, it does not examine its consequence on patient outcomes. Although this potential analysis is mentioned in future directions, it is a necessary piece to demonstrate substantial enough significance for publication in this journal.

In this study, we focused more on behavioral change under AI-assistance. To directly validate the AI utility, a future prospective study would be necessary. Nevertheless, we attempted to evaluate AI utility indirectly on the basis of projected outcomes (tcp , $ntcp$, Figures 7 and 8). We scored each adjusted decision using $tcp(1 - ntcp)$ function, which ranges from 0 to 1, where 1 corresponds to toxicity-free local control ($tcp = 1, ntcp = 0$). When we analyzed the distribution of the scores, we found positively skewed scores, indicating an overall benefit. We have now highlighted that in the Discussion.

5. One key element missing from experimental design is whether or not the treating radiation oncologist is likely to change their case approach if and when AICDSS provides suggestions that are far afield from reasonable. By 'spiking' in bad suggestions purposely, are the radiation oncologists editing the case still likely to change their approach in the direction of the AICDSS recommendation? If the model performs poorly, how likely is the treating physician to in turn take an approach that could lead to patient harm? This is a crucial element and a great fear of over-reliance on AI that this study could have investigated but we are left guessing - possibly?

We thank the reviewer for raising a crucial point which have been raised by other reviewers. In our case, the main difficulty was to objectively classify AI's recommendation as good or bad. This is due to a lack of ground truth and also the gradual nature (continuous nature) of the decision with respect to treatment outcome. We have acknowledged the importance of the suggested experiment in the limitation and future studies. On the other hand, we believe that increasing model transparency and interpretability will help mitigate the risk associated with faulty AI or even bad actors. In our study, we found that evaluators would override AI's recommendation if they found it harmful and make their own judgment based on the projected outcome.

Reviewer #6 (Remarks to the Author):

We thank the reviewer for their time.

Reviewer #6 (Remarks on code availability):

The authors provided access to code on GitHub and the ARClIDS user interface, which works well enough to demonstrate the AI system functions. I did not personally attempt to install and run the code on my local device.

We thank the reviewer for the positive comment.